# A TMPRSS2 inhibitor acts as a pan-SARS-CoV-2 prophylactic and therapeutic

Tirosh Shapira[1,7], I. Abrrey Monreal[2,7], Sébastien P. Dion[3], David W. Buchholz[2], Brian Imbiakha[2], Andrea D. Olmstead[1], Mason Jager[2], Antoine Désilets[3], Guang Gao[1,4], Mathias Martins[5], Thierry Vandal[3], Connor A. H. Thompson[1], Aaleigha Chin[1], William D. Rees[6], Theodore Steiner[6], Ivan Robert Nabi[4], Eric Marsault[3,8], Julie Sahler[2], Diego G. Diel[5], Gerlinde R. Van de Walle[2], Avery August[2], Gary R. Whittaker[2], Pierre-Luc Boudreault[3], Richard Leduc[3✉], Hector C. Aguilar[2✉] & François Jean[1✉]

The COVID-19 pandemic caused by the SARS-CoV-2 virus remains a global public health crisis. Although widespread vaccination campaigns are underway, their efficacy is reduced owing to emerging variants of concern[1,2]. Development of host-directed therapeutics and prophylactics could limit such resistance and offer urgently needed protection against variants of concern[3,4]. Attractive pharmacological targets to impede viral entry include type-II transmembrane serine proteases (TTSPs) such as TMPRSS2; these proteases cleave the viral spike protein to expose the fusion peptide for cell entry, and thus have an essential role in the virus lifecycle[5,6]. Here we identify and characterize a small-molecule compound, N-0385, which exhibits low nanomolar potency and a selectivity index of higher than $10^6$ in inhibiting SARS-CoV-2 infection in human lung cells and in donor-derived colonoids[7]. In Calu-3 cells it inhibits the entry of the SARS-CoV-2 variants of concern B.1.1.7 (Alpha), B.1.351 (Beta), P.1 (Gamma) and B.1.617.2 (Delta). Notably, in the K18-human ACE2 transgenic mouse model of severe COVID-19, we found that N-0385 affords a high level of prophylactic and therapeutic benefit after multiple administrations or even after a single administration. Together, our findings show that TTSP-mediated proteolytic maturation of the spike protein is critical for SARS-CoV-2 infection in vivo, and suggest that N-0385 provides an effective early treatment option against COVID-19 and emerging SARS-CoV-2 variants of concern.

In December 2019, the first cases of COVID-19 emerged in Wuhan, Hubei Province, China, and were rapidly attributed to the aetiology of a novel β-coronavirus, SARS-CoV-2[8]. As of 18 January 2022, more than 332 million SARS-CoV-2 infections and 5.6 million deaths have been reported[9]. The approval and widespread distribution of several highly effective vaccines, along with other public health measures, has been instrumental in controlling the COVID-19 pandemic; however, novel genetic variants of SARS-CoV-2 are emerging and spreading at an alarming rate[10]. Notably, vaccine effectiveness may be reduced against a number of these variants, termed variants of concern (VOCs), including B.1.1.7, P.1, B.1.351 and B.1.617.2[2,11]. In particular, the recent emergence of a novel, heavily mutated VOC, B.1.1.529[1,12] has shown that the SARS-CoV-2 pandemic is likely to remain a global health threat for the foreseeable future.

Discovering novel classes of antiviral compounds—including both direct-acting antivirals (DAAs) and host-directed antivirals (HDAs)—and intensive in cellulo and in vivo studies of their antiviral profiles as mono- or combination therapies against emerging SARS-CoV-2 VOCs are critical for developing preventive and therapeutic strategies to combat COVID-19[13,14]. At present, three antivirals have been approved for clinical use against SARS-CoV-2. Remdesivir is a DAA that targets the viral RNA-dependent RNA polymerase that catalyses the synthesis of viral RNA[15]. Remdesivir is administered intravenously to hospitalized individuals with COVID-19[16]. Other DAAs, PF-07321332 (paxlovid) and MK-4482/EIDD-2801 (molnupiravir), have also been developed as oral clinical candidates[17,18]. Paxlovid targets the main protease of the coronavirus (3CL^pro, also known as M^pro), an essential protease that is involved in processing viral replicase polyproteins, whereas molnupiravir is a ribonucleoside analogue that inhibits viral

[1]Department of Microbiology and Immunology, Life Sciences Institute, University of British Columbia, Vancouver, British Columbia, Canada. [2]Department of Microbiology and Immunology, Cornell University College of Veterinary Medicine, Ithaca, NY, USA. [3]Department of Pharmacology-Physiology, Faculty of Medicine and Health Sciences, Institut de Pharmacologie de Sherbrooke, Université de Sherbrooke, Sherbrooke, Québec, Canada. [4]Department of Cellular and Physiological Sciences, Life Sciences Institute, University of British Columbia, Vancouver, British Columbia, Canada. [5]Department of Population Medicine and Diagnostic Sciences, Cornell University College of Veterinary Medicine, Ithaca, NY, USA. [6]Department of Medicine, BC Children's Hospital Research Institute, University of British Columbia, Vancouver, British Columbia, Canada. [7]These authors contributed equally: Tirosh Shapira, I. Abrrey Monreal. [8]Deceased: Eric Marsault. ✉e-mail: richard.leduc@usherbrooke.ca; ha363@cornell.edu; fjean@mail.ubc.ca

replication[17,18]. Alternatively, HDAs (also termed indirect-acting antivirals) are under investigation and may offer a complement to DAAs. Emerging SARS-CoV-2 VOCs are less likely to develop resistance to HDAs than to DAAs because, unlike viral genes, host genes have a low propensity to mutate[5,13]. Camostat mesylate (Cm), for example, is a broad spectrum serine protease inhibitor used to treat pancreatitis that has been repositioned as a clinical candidate for treating COVID-19[4,5,19].

Accumulating evidence has shown that SARS-CoV-2 is dependent on host pathways, including the hijacking of TMPRSS2-related proteases for viral entry; this suggests that TTSPs could be therapeutic targets to prevent SARS-CoV-2 infection[5,6,20]. The SARS-CoV-2 lifecycle begins with attachment and entry into respiratory epithelium via the angiotensin-converting enzyme 2 (ACE2) receptor[4,8]. This is mediated by the major viral surface glycoprotein, spike (S), which must undergo two sequential proteolytic cleavages by host proteases before it can mediate fusion of the virus with host cell membranes, a requirement for subsequent viral replication[3,21,22]. The first spike cleavage occurs at the S1/S2 site, releasing S1 and S2 subunits that remain non-covalently linked; this event is likely to be mediated by host furin-like proteases[21,22]. The second cleavage occurs at the S2′ site, immediately adjacent to the fusion peptide. This cleavage, which triggers the fusion event, is mediated by host TTSPs such as TMPRSS2 and TMPRSS13, which cleave after specific single arginine or lysine residues[4,23].

The K18-hACE2 mouse model (transgenic expression of human ACE2 (hACE2) under a cytokeratin 18 promoter) offers a stringent system for testing the efficacy of DAAs and HDAs against severe disease and mortality after SARS-CoV-2 infection[24]. So far, only a few studies have tested antiviral efficacy in this animal model, with only one DAA reported as protective against lethal SARS-CoV-2 infection in this model[25,26].

Here we report on the design and testing of peptidomimetics for their inhibitory activity against TMPRSS2 and related TTSPs. We then investigated the antiviral activities of the peptidomimetics against an ancestral strain of SARS-CoV-2 (lineage B, VIDO) and four variants—B.1.1.7 (Alpha), B.1.351 (Beta), P.1 (Gamma) and B.1.617.2 (Delta)—in human lung cells. Finally, we tested our top highly potent antiviral, N-0385, against SARS-CoV-2 (lineage A strain) and SARS-CoV-2 Delta-induced morbidity and mortality in K18-hACE2 mice. We found that N-0385 provides a high level of protection and a therapeutic benefit after either multiple administrations or a single administration in this model of severe disease. Thus, N-0385 is an antiviral with a high potential for use against COVID-19.

## Peptidomimetics potently inhibit TMPRSS2

We previously designed first-generation peptidomimetic tetrapeptide compounds with ketobenzothiazole warheads, and they exhibited inhibitory activity against a host TTSP, matriptase[27,28]. These compounds act as slow tight-binding inhibitors in vitro but their potency in cellular systems was modest against influenza A virus[28]. To improve their stability and potency, we modified their N terminus either by capping or through the synthesis of desamino moieties[29] (Fig. 1a, Extended Data Fig. 1a). When we measured the stability of the desamino compounds, we found that they had markedly increased half-lives compared to their corresponding amine analogues (48 h versus 2 h, respectively, in human lung epithelial Calu-3 cells) (data not shown). Moreover, these compounds exhibited low nanomolar efficacies when tested in H1N1 models of influenza A virus infection[28,30].

Expanding on that work here, we developed a small library of peptidomimetic compounds (Fig. 1a, Extended Data Fig. 1a) to screen for inhibition of TMPRSS2 proteolytic activity, as this TTSP is a crucial host protease that is involved in cleaving the SARS-CoV-2 spike protein and priming the virus for cell entry[4]. We included in this screen our first-generation tetrapeptide[28], N-0100, which lacks an N-terminal stabilizing group, along with three desamino tetrapeptide analogues. We also tested four tripeptides containing different N-terminal capping groups.

To evaluate the efficacies of these compounds, we set up a cellular assay to measure TMPRSS2-dependent pericellular inhibition of proteolytic activity. We expressed the full-length, wild-type TMPRSS2 or an inactive form of the protease in which the serine residue of the catalytic triad was replaced by alanine (TMPRSS2(S441A)) in Vero E6 cells. Twenty-four hours after transfection, the medium was replaced for an additional 24 h with serum-free medium containing vehicle or compound in the presence of a TMPRSS2-preferred fluorogenic substrate[31] (Fig. 1b). Using this assay, we show that, as expected, the S441A substitution completely abrogated the proteolytic activity of TMPRSS2.

The peptidomimetics were initially tested for inhibitory activity against human TMPRSS2 at 10 nM (Fig. 1b). Treatment with Cm, which has previously been shown to be active against TMPRSS2[23], reduced substrate proteolysis by 56% compared to untreated TMPRSS2-expressing cells. The first-generation peptidomimetic, N-0100, did not inhibit TMPRSS2 activity under these conditions. However, the more stable tetrapeptides with N-terminal desamino moieties, N-0130 and N-0438, had increased inhibitory activities of 72% and 84%, respectively. N-0678 (substituting P2 Phe for the synthetic amino acid cyclohexylalanine (Cha)) only inhibited TMPRSS2 activity by 5%. N-0676 (a tripeptide with an N-terminal acetyl (Ac) cap and P2 Cha) also weakly inhibited TMPRSS2 activity by 8%. N-0386 (with restored Phe in P2) resulted in a more potent inhibition of 73%. N-1296 (replacing Ac with amidinyl (Am)) had reduced potency of 16%, whereas N-0385 (replacing Am with mesyl (Ms)) resulted in a highly potent inhibition of 83%. Notably, several peptidomimetic compounds were more efficient than Cm at reducing the activity of TMPRSS2 (Fig. 1b).

We then investigated the dose response of the four most promising peptidomimetics (N-0130, N-0385, N-0386 and N-0438). The half-maximal inhibitory concentration (IC$_{50}$) of Cm was $17.5 \pm 18.8$ nM, whereas the IC$_{50}$ values for the peptidomimetics were $3.1 \pm 1.5$ nM (N-0130), $5.2 \pm 5.4$ nM (N-0438), $3.9 \pm 4.4$ nM (N-0386) and $1.9 \pm 1.4$ nM (N-0385) (Fig. 1c, Extended Data Table 1). Of note, none of the compounds affected the cellular viability of Vero E6 cells when used at 10 µM (Extended Data Fig. 1b). To confirm the contribution of the ketobenzothiazole warhead to the inhibitory activity of the molecule, the ketone functional group of N-0385 was replaced with an alcohol group to generate N-0385(OH) (Fig. 1a), which we expected would no longer trap the target protease. As expected, no significant reduction in TMPRSS2 activity was detected when cells were treated with up to 10 µM of N-0385(OH) (Fig. 1c), suggesting that the integrity of the ketobenzothiazole group is required to achieve potency. We also confirmed the efficacy of N-0385 against mouse TMPRSS2 with an IC$_{50}$ of $12.3 \pm 1.9$ nM (Extended Data Fig. 1c). Next, we sought to determine the selectivity profile of these inhibitors by measuring the inhibition constant $K_i$ on selected recombinant serine proteases, including three members of the TTSP family (matriptase, hepsin and DESC1) as well as furin, thrombin, and cathepsin L. All four of the tested peptidomimetic compounds behaved as low nanomolar inhibitors for the TTSPs, but they were inactive or showed only weak inhibition against the other proteases (Fig. 1d, Extended Data Table 2). Cm had a similar selectivity profile to the peptidomimetics tested, except that it showed moderate inhibition of thrombin ($K_i = 621$ nM), in line with its broader-spectrum properties. Overall, these data show that TTSP-targeting peptidomimetics containing a ketobenzothiazole warhead inhibit TMPRSS2-dependent pericellular activity in a cellular assay and preferentially inhibit other members of the TTSP family.

To understand the mode of binding and the main interactions of our inhibitors and how these compounds achieve their high inhibitory potential, we built a homology model of TMPRSS2 using the crystal structure of matriptase (Protein Data Bank (PDB): 6N4T)[32]. Alignment of the catalytic domains revealed 41% and 60% identity and sequence similarity, respectively, making it a reliable model, especially near the conserved binding site. Docking of N-0385 was modelled to this structure (Fig. 1e). As predicted and previously published[32], the catalytic

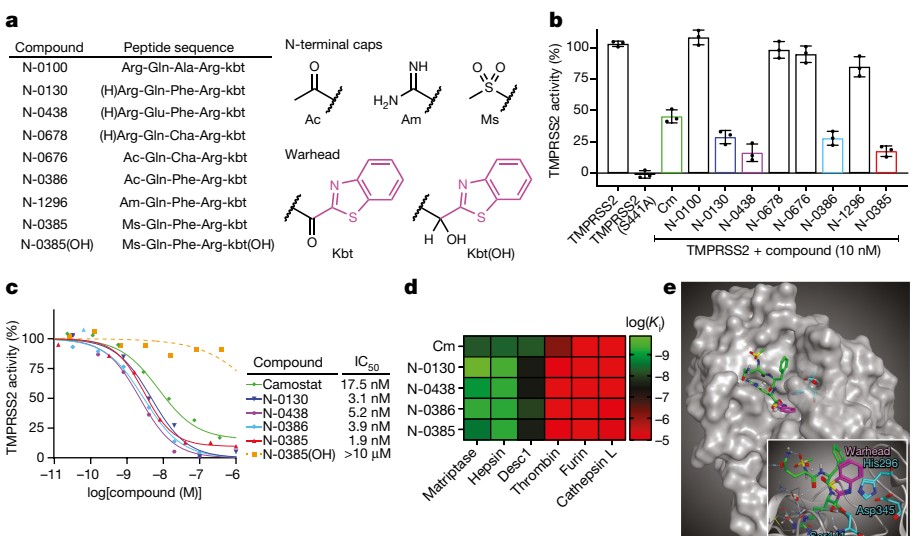

**Fig. 1 | Ketobenzothiazole-based peptidomimetics are potent TMPRSS2 inhibitors. a**, Peptidomimetic compounds used in this study along with their respective sequences. The structures of N-terminal caps, the ketobenzothiazole warhead and the alcohol ketobenzothiazole are shown on the right. (H)Arg, desamino arginine; kbt, ketobenzothiazol. **b**, Vero E6 cells were transfected with an empty vector (mock), wild-type TMPRSS2 or the inactive mutant TMPRSS2(S441A) for 24 h. The indicated compounds (10 nM) were added concomitantly with a fluorogenic substrate on cells for an additional 24 h before fluorescence reading. Relative TMPRSS2 activity was measured using the mock-subtracted fluorescence and is reported as the percentage of residual activity relative to the vehicle-treated cells (0.01% DMSO). Data are presented as mean ± s.d. ($n = 3$ independent experiments). **c**,

Dose–response curves were generated for the indicated compounds using the assay described in **b**, and $IC_{50}$ values were determined using nonlinear regression analysis. Representative $IC_{50}$ curves are shown, with the mean value of independent experiments ($n = 3$ for N-0130, N-0386 and N-0385(OH); $n = 4$ for Cm and N-0385; $n = 5$ for N-0438). **d**, Specificity of selected compounds toward other serine proteases. Data are the mean of $\log(K_i)$; $n = 3$ independent experiments (except cathepsin L versus N-0385, $n = 4$) and are shown as a heat map. **e**, Main image, docking of N-0385 (green; warhead in purple) in the binding pocket of TMPRSS2 (homology model). Residues of the catalytic triad are shown in cyan. Inset, interaction of N-0385 with TMPRSS2 residues. N-0385 forms a covalent bond with the catalytic triad residue Ser441.

triad Ser441 (catalytic triad: Ser441, His296, and Asp345; Fig. 1e) forms a covalent bond with the warhead ketone, thus leading to a tight-binding mode of inhibition.

Several key interactions can be observed in the binding pocket. As in all TTSP inhibitors possessing a guanidine group on the sidechain, a strong hydrogen bond network stabilizes this pharmacophore deep within the binding pocket (Fig. 1e). This includes Asp435 and Gly464, as well as Gln438 via a water molecule. Gln438 is also involved in another hydrogen bond of this same water molecule to the oxygen of the main-chain ketone group. This ketone also acts as a hydrogen bond acceptor with Gly462. The N-terminal mesylate forms two hydrogen bonds—one intramolecular with the side-chain amide of the Gln residue of N-0385, and another with Gly462. Finally, the oxygen of the newly formed hemiacetal is stabilized by two hydrogen bond donors from the Gly439 and Ser441 amines. A portion of the ketobenzothiazole warhead and the aromatic ring from the phenylalanine are exposed to the solvents, which could allow us to further optimize the design of this second-generation inhibitor, leading to an improved pharmacokinetic profile.

## N-0385 inhibits SARS-CoV-2 infection

The peptidomimetic compounds that we screened against TMPRSS2 were subsequently tested for their efficacy at preventing SARS-CoV-2 infection. Calu-3 cells were pretreated with 100 nM of the compounds for 3 h before infection with an ancestral SARS-CoV-2 lineage B strain (VIDO). Cells were fixed and immunofluorescently stained for double-stranded RNA (dsRNA), a marker of viral replication[33], and for the viral nucleocapsid, a marker of viral entry and translation[6] (Extended Data Fig. 2). Fluorescent high-content imaging and relative quantification of virally infected cells showed consistent inhibitory profiles across dsRNA and nucleocapsid staining, which mirrored the

inhibitory profile observed in the TMPRSS2 proteolytic activity assay (Fig. 2a versus Fig. 1b). Cm, which interferes with SARS-CoV-2 infection[4], reduced infection by more than 83% compared to non-treated cells. N-0100, which lacks an N-terminal stabilizing moiety, reduced infection by less than 25%. The tetrapeptides N-0130 and N-0438, which have N-terminus desamino moieties, had greatly increased antiviral activity of more than 93% and more than 88%, respectively. N-0678 (substituting P2 Phe for the synthetic amino acid Cha) inhibited SARS-CoV-2 by less than 23%. N-0676 (tripeptide with an N-terminal Ac cap and P2 Cha) had only moderate inhibitory activity of less than 53%. N-0386 (with restored Phe in P2) resulted in a highly potent SARS-CoV-2 inhibition of greater than 99%. N-1296 (replacing Ac with Am) reduced the antiviral potency to less than 44%, whereas N-0385 (capped with Ms) restored antiviral activity to over 99%. Finally, N-0385(OH) (with OH replacing the functional group of the warhead), showed an inhibition of SARS-CoV-2 of less than 23%. Thus, TMPRSS2-inhibiting peptidomimetics are also inhibitors of SARS-CoV-2 replication and translation in Calu-3 cells, and the stabilizing N-terminal caps and the ketobenzothiazole warhead are likely to be essential for compound stability and antiviral potency.

Compounds that inhibited SARS-CoV-2 (inhibitory activity of greater than 75%) in the antiviral screen were further validated and characterized using a dose–response analysis in Calu-3 cells (Fig. 3). The half-maximal effective concentration ($EC_{50}$) of Cm was $10.6 ± 8.4$ nM, whereas the $EC_{50}$ values for the other compounds were $30.1 ± 30.1$ nM (N-0130), $35.7 ± 24.5$ nM (N-0438), $2.3 ± 1.7$ nM (N-0386) and $2.8 ± 1.4$ nM (N-0385) (Fig. 2b). An $EC_{50}$ value could not be determined for N-0385(OH) as substantial inhibition was not observed at concentrations up to 1 μM (Fig. 2b, Extended Data Fig. 2). These compounds did not exhibit any toxicity; all four compounds had half-maximal cytotoxic concentration ($CC_{50}$) values of 1 mM or greater in Calu-3 cells (Extended Data Fig. 3, Extended Data Table 1). Thus, the selectivity

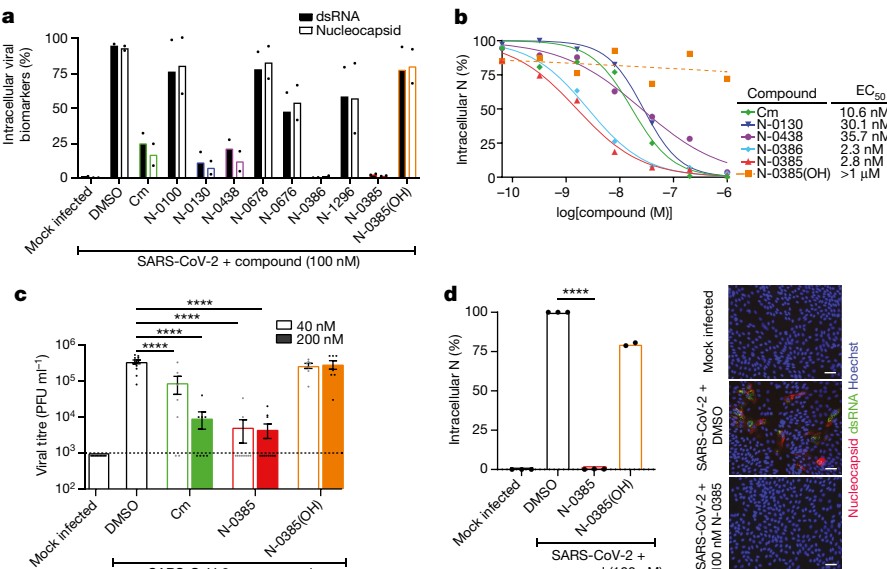

**Fig. 2 | Peptidomimetics active against TMPRSS2 are potent low nanomolar inhibitors of SARS-CoV-2 in a human lung epithelial cell line and in human colonoids. a**, Calu-3 cells were pretreated with 100 nM of the indicated compounds followed by SARS-CoV-2 (VIDO) infection (multiplicity of infection (MOI) = 2)). Intracellular infection levels were evaluated by high-content screening of cell nuclei, dsRNA and nucleocapsid and then quantified relative to DMSO-treated cells ($n = 2$ independent experiments). **b**, Dose–response curves were generated for the lead antiviral peptidomimetic compounds in Calu-3 cells using nucleocapsid (N) staining of cells that were pretreated with the indicated compounds before infection (Cm, $n = 5$; N-0130, $n = 5$; N-0438, $n = 3$; N-0386, $n = 4$; N-0385, $n = 8$; N-0385(OH), $n = 5$). **c**, Plaque assays were performed using two of the experimental conditions evaluated in the dose–response analysis (40 nM and 200 nM) to determine the viral titres (amount of infectious virus) produced in cells that were pretreated with the indicated compounds before infection ($n = 3$ independent experiments); dotted line represents limit of detection. **d**, Colonoids were pretreated with 100 nM of the indicated compounds and infected with SARS-CoV-2 (MOI ≈ 1). Intracellular infection was relatively quantified using N staining. (N-0385, $n = 3$; N-0385(OH), $n = 2$). Representative fluorescent images of colonoids subjected to the indicated treatments are shown (Hoechst in blue, nucleocapsid in red and dsRNA in green). Scale bars, 50 μm. One-way ANOVA with Bonferroni correction was used to determine significance in **c**, **d**; **** indicates modified $P < 0.0001$. Error bars, s.e.m.

index for these compounds (N-0130, N-0438, N-0386 and N-0385) was between $8.97 × 10^4$ and $2.75 × 10^6$ (Extended Data Table 1). Overall, these results confirm that two TTSP-targeted peptidomimetic compounds (N-0386 and N-0385) are extremely potent low nanomolar inhibitors of SARS-CoV-2 infection in human lung epithelial cells.

We next examined the effects of Cm, N-0385 and N-0385(OH) on the extracellular release of SARS-CoV-2 infectious virions from Calu-3 cells. Two effective doses (40 nM and 200 nM) from the $EC_{50}$ curve-fitting (Fig. 2b) were selected for plaque assays. The cell supernatant from Cm-treated and SARS-CoV-2-infected cells showed an approximately half-log reduction of viral titre in the presence of 40 nM Cm compared to the DMSO-treated and infected control and an approximately 1.5-log reduction with 200 nM Cm (Fig. 2c). In comparison, both 40 nM and 200 nM treatments with N-0385 reduced viral titres by almost 2-log. Consistent with previous results, N-0385(OH) did not induce reduction in SARS-CoV-2 plaques at 40 nM or 200 nM. These results confirm that N-0385, which targets TMPRSS2, is a potent inhibitor of SARS-CoV-2 infectivity in Calu-3 cells and that the ketobenzothiazole warhead is required for N-0385 antiviral potency.

Although Calu-3 cells represent a scalable and clinically relevant system of antiviral screening for SARS-CoV-2 inhibitors, they are an immortalized cell line. To evaluate the effectiveness of N-0385 in a primary human cell-based model, we examined SARS-CoV-2 infection in donor-derived human colonoids[7,34]. SARS-CoV-2 initially causes a respiratory infection, but many infected individuals also experience gastrointestinal symptoms that are frequently linked with increased disease duration and severity[35]. A recent report identified TMPRSS2 as an essential co-factor for SARS-CoV-2 infection in colonoids[20]. We first relatively quantified the mRNA expression of *ACE2* and *TMPRSS2* in colonoids and Calu-3 cells using quantitative PCR (qPCR). *ACE2* showed comparable levels of expression in colonoids compared to

Calu-3 cells, whereas *TMPRSS2* had much higher expression levels in colonoids compared to Calu-3 cells (Extended Data Fig. 1d, e). We then investigated the susceptibility of colonoid monolayers to SARS-CoV-2 infection. Consistent with previous work, the colonoids were susceptible to infection, as evidenced by dsRNA and nucleocapsid staining (Fig. 2d, Extended Data Fig. 4).

N-0385 and N-0385(OH) were then tested for their efficacy at preventing SARS-CoV-2 infection in colonoids. The colonoids were pretreated with 100 nM of the compounds for 3 h before being infected with SARS-CoV-2 for 3 days. Under these conditions, infection was undetectable in colonoids that were pretreated with N-0385 (greater than 99% inhibition), when compared with DMSO-treated colonoids (Fig. 2d). By contrast, N-0385(OH) did not significantly reduce SARS-CoV-2 infection in this system (less than 20% inhibition) (Fig. 2d). These results align with observations in Calu-3 cells and confirm the nanomolar potency of N-0385 against SARS-CoV-2 in primary human cells.

## N-0385 inhibits infection with SARS-CoV-2 VOCs

To our knowledge, mutations in the TMPRSS2 cleavage site have not been identified in SARS-CoV-2 variants, which suggests that N-0385 should retain high potency against SARS-CoV-2 VOCs[11]. First, we confirmed the infectivity of four VOCs in Calu-3 cells: B.1.1.7 (Alpha), B.1.351 (Beta), P.1 (Gamma) and B.1.617.2 (Delta). Confocal imaging of infected cells confirmed the infectivity of these variants, as demonstrated by nucleocapsid and dsRNA staining (Fig. 3a). Although the viral marker staining patterns were relatively consistent in Calu-3 cells infected with a lineage B isolate (VIDO), B.1.1.7, B.1.351 and P.1, we observed a striking spheroid-like phenotype in cells infected with B.1.617.2 (Fig. 3a, b, Supplementary Video 1). We then evaluated the efficacy of N-0385 for preventing infection with SARS-CoV-2 VOCs in Calu-3 cells. The $EC_{50}$

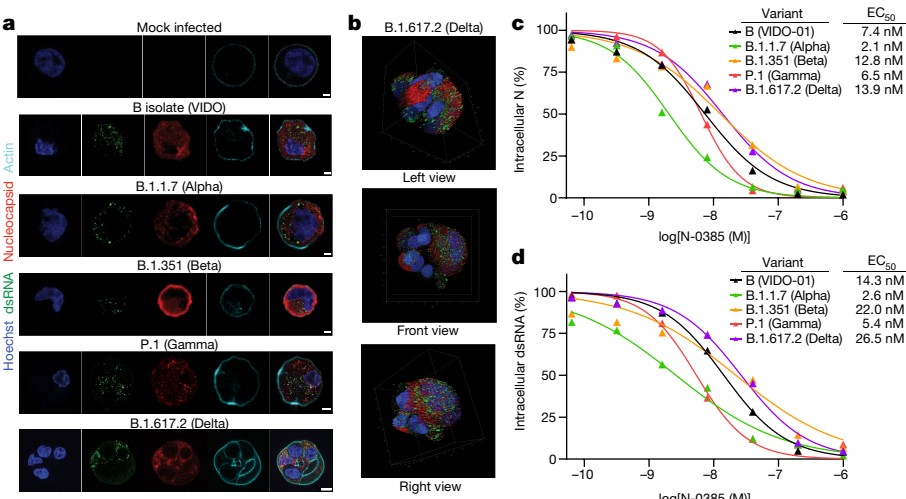

**Fig. 3 | N-0385 is a nanomolar inhibitor of four SARS-CoV-2 VOCs. a**, Representative fluorescent images of SARS-CoV-2-infected Calu-3 cells. Calu-3 cells infected with the indicated SARS-CoV-2 variants and mock infected are shown. Scale bars, 5 µm. **b**, Representative images from a 3D volume rendering of Delta-infected cells. In **a**, **b**, Hoechst is shown in blue, nucleocapsid (N) in red, dsRNA in green and actin in cyan; images were captured with a Leica TCS SP8 3× STED microscope. **c**, **d**, Dose–response curves were generated for N-0385 in Calu-3 cells using N staining (**c**) and dsRNA staining (**d**) of N-0385 pretreated cells infected with the indicated VOCs ($n = 4$ independent experiments).

of N-0385 against all VOCs was in the low nanomolar range, ranging from 2.1 nM to 13.9 nM using nucleocapsid staining as a marker, and ranging from 2.6 nM to 26.5 nM using dsRNA staining as a marker of infection (Fig. 3c, d). This underscores the potential of N-0385 to act as a pan-variant, host-directed antiviral against emerging SARS-CoV-2 VOCs.

## N-0385 protects mice against COVID-19

After establishing the efficacy of N-0385 in vitro and in cellulo, we tested whether intranasal administration would improve morbidity and survival in vivo, using K18-hACE2 mice[36,37], an established mouse model of severe COVID-19[38]. Dosing regimens and drug concentrations were chosen on the basis of preliminary studies performed in a mouse model of influenza A virus infection, which showed antiviral efficacy at 7.2 mg kg[-1] (data not shown), and also on the basis of the solubility of N-0385 and on the knowledge that K18-hACE2 mice typically survive 6 to 8 days post-infection (dpi) with SARS-CoV-2[39]. Ten mice per group (five females and five males) were administered a single daily intranasal dose of 7.2 mg kg[-1] N-0385, N-0385(OH) or a vehicle control (0.9% saline) for eight days from day −1 to day 6 relative to infection. The mice were challenged on day 0 with $1 \times 10^3$ plaque-forming units (PFU) of SARS-CoV-2 per mouse, and surviving mice were monitored until the study end-point (14 dpi) (Fig. 4a). At 6 dpi (before any mice had died) the mice that were treated with saline control and N-0385(OH) lost on average 14% and 12% of their weight, respectively, whereas N-0385-treated mice lost on average only 3% of their weight. The relative changes in weight were maintained when compared at the study end-point (14%, 15% and 3% weight loss for mice treated with saline, N-0385(OH) and N-0385, respectively) (Fig. 4b–e). As expected, most of the saline- and N-0385(OH)-treated mice died at 6–9 dpi, with 0% and 10% surviving to the end-point, respectively. By contrast, 70% of the N-0385-treated mice survived to the end-point (Fig. 4f).

In this in vivo experiment, histological examination of lung tissue obtained either at the time of death (6–9 dpi) or the study end-point (14 dpi) revealed mild pathology in most SARS-CoV-2 infected mice, with mild perivascular and interstitial inflammatory infiltrates as the predominant change, irrespective of the treatment group (Extended Data Table 3). Compared to N-0385-treated mice, control saline-treated mice frequently had additional histological changes including alveolar oedema, alveolar fibrin and inflammatory cells within alveoli. Of the mice that survived up to the study end-point, three had focal areas of fibrosis, type II pneumocyte hyperplasia and occasionally lymphoid hyperplasia. However, most of the mice that survived showed little to no pathological signs in the lungs (Fig. 4g). Histological lesions in the brain included multifocal perivascular cuffs of inflammatory cells, reactive glial cells, neutrophils and lymphocytes in the adjacent neuroparenchyma (gliosis), infiltration of the meninges with inflammatory cells, and neuronal necrosis characterized by shrunken neuron bodies with hypereosinophilic cytoplasm and pyknotic or karyorrhectic nuclei. No lesions were observed in the brains of mice that survived to the study end-point (Fig. 4h, Extended Data Table 3). Immunohistochemistry (IHC) of the SARS-CoV-2 nucleocapsid protein and plaque assay from the tissues collected at the time of death (6–9 dpi) or the study end-point (14 dpi) revealed large amounts of the viral antigen and high viral titres in the tissues of infected mice treated with saline or N-0385(OH) (Extended Data Fig. 5, Extended Data Table 4). Although samples obtained at different time points are not directly comparable, the amounts of antigen and viral titres were lower in mice treated with N-0385, particularly in those that survived to the study end-point (Extended Data Fig. 5).

After establishing that N-0385 improves the survival of SARS-CoV-2 infected mice, we evaluated the outcome of a shortened treatment regimen (4 days) on the survival of K18-hACE2 mice (Fig. 5). Furthermore, to accurately compare viral loads after infection, we also analysed viral lung titres and IHC at the same timepoint (3 dpi) in equivalently infected mice groups, in addition to analysing titres at the time of death or the study end-point (14 dpi). Five female and five male mice per group were treated with N-0385 or saline from day −1 to day 2 relative to infection (Fig. 5a). The N-0385-treated mice in this group showed 100% survival compared to 20% survival in the control group (Fig. 5b). At 6 dpi (before any mice had died) the saline control mice lost on average 10% of their weight, whereas N-0385 treated mice gained 1% weight. When compared at time of death or the study end-point, the saline control mice lost an average of 14% of their weight, whereas N-0385 treated mice gained 2% weight (Fig. 5c–e). Our analysis of viral loads through plaque assays and IHC at 3 dpi showed that N-0385-treated mice had significantly reduced viral titres (97%) and IHC staining (98%) compared to control saline-treated mice (Fig. 5f, g). No infectious virus was detected in the lung of N-0385-treated mice at the study end-point, or

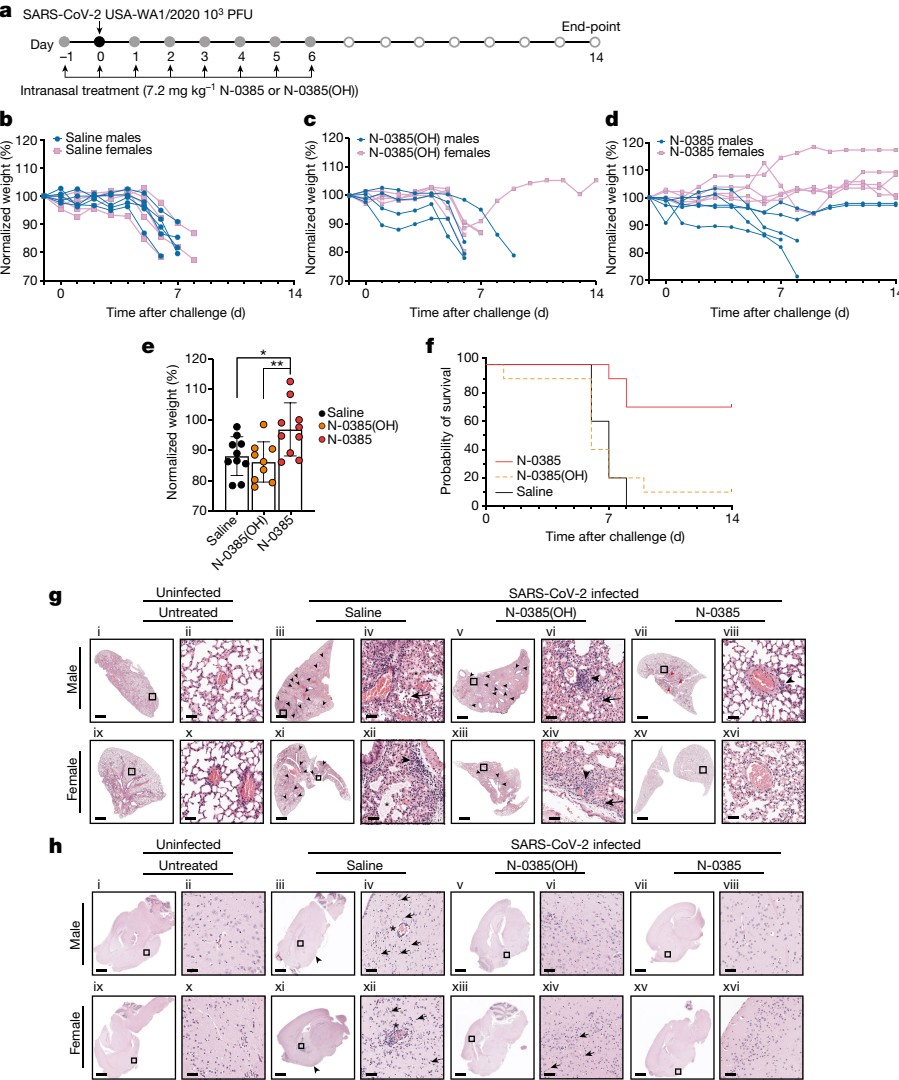

**Fig. 4 | N-0385 reduces morbidity and mortality in the K18-hACE2 mouse model of SARS-CoV-2. a**, Mice (*n* = 10 per treatment group) were treated daily on days −1 to +6 relative to infection. Surviving mice were euthanized at the study end-point. **b**–**d**, Weight change of mice treated with saline (control) (**b**), N-0385(OH) (**c**) or N-0385 (**d**). **e**, Weight loss difference at 6 dpi; *P = 0.0193, **P = 0.0083. Data are mean ± s.d., with two-tailed Student's *t*-tests used to determine significance. **f**, Probability of survival. **g**, Representative haematoxylin and eosin (H&E) staining of lung histopathology (one slice analysed per mouse) at death or study end-point in untreated mice (i, ii, ix, x) or mice treated with saline (iii, iv, day 7; xi, xii, day 6), N-0385(OH) (v, vi, xiii, xiv, day 6) or N-0385 (vii, viii, xv, xvi, day 14). Uninfected mice tissues (i, ii, ix, x) were normal. Challenged mice (iii–viii, xi–xvi) developed perivascular infiltrates of inflammatory cells (arrowheads). Severe inflammatory changes including alveolar fibrin and oedema (asterisks) were found only in the saline group

(iii, iv, xi, xii). Perivascular inflammatory cell infiltrates (arrowheads) were more widespread in saline (iii, xi) and control N-0385(OH) (v, xiii) compared to N-0385 mice (vii, xv). Surviving N-0385 mice (vii, viii, xv, xvi) had smaller and fewer perivascular inflammatory infiltrates (arrowheads) and occasional type II pneumocyte hyperplasia (red arrows). **h**, Representative H&E images of brain histopathology (one slice analysed per mouse) in untreated mice (i, ii, ix, x) or mice treated with saline (iii, iv, day 7; xi, xii, day 8), N-0385(OH) (v, vi, day 6; xiii, xiv, day 7) or N-0385 (vii, viii, xv, xvi, day 14). Saline-treated mice (i, ii, ix, x) developed perivascular cuffs of inflammatory cells (asterisks), necrotic neurons (arrows), gliosis and meningeal infiltrates (arrowheads). Brain lesions were reduced in N-0385(OH) mice (v, vi, xiii, xiv) and absent in surviving N-0385 mice (vii, viii, xv, xvi). The magnified areas were selected to best represent the presence of inflammatory cells and pathological changes. Scale bars in **g**, **h**: i, iii, v, vii, ix, xi, xiii, xv, 1 mm; ii, iv, vi, viii, x, xii, xiv, xvi, 50 µm.

in the two saline-treated mice that survived to the end-point. (Fig. 5h). This demonstrates the effectiveness of N-0385 in blocking SARS-CoV-2 infection and improving disease outcomes and survival using a short, early treatment regimen.

## N-0385 protects mice against the Delta VOC

Next, we further investigated the treatment window of N-0385 as well as the pan-variant effectiveness against SARS-CoV-2 B.1.617.2 in mice using single doses of N-0385. Mice (5 males and 5 females per group) were challenged on day 0 with 1 × 10³ PFU of SARS-CoV-2 B.1.617.2 per mouse (Fig. 6). Mice that were administered a single intranasal dose of

saline at 12 h post-infection (hpi) (Fig. 6a) were compared to mice that were treated with N-0385 at 12 hpi (N-0385 12 hpi, 14.4 mg kg⁻¹) (Fig. 6b) or at the time of infection (N-0385 0 hpi, 7.2 mg kg⁻¹) (Fig. 6c). Weight was monitored for six days after infection. N-0385 showed significant protection against infection-associated weight loss (Fig. 6d); the lowest weight loss occurred when N-0385 was administered at the time of infection (0 hpi) (2% weight gain versus 13% weight loss in control mice) (Fig. 6c, d). Protection also occurred when mice were treated at 12 hpi with N-0385 (5% weight loss versus 14% weight loss in control mice) (Fig. 6b, d). In a similar experiment, mice were treated with N-0385 or saline control at 12 hpi and lung tissue was collected at 3 dpi for plaque assays and IHC to measure viral titres and nucleocapsid staining (Fig. 6e, f).

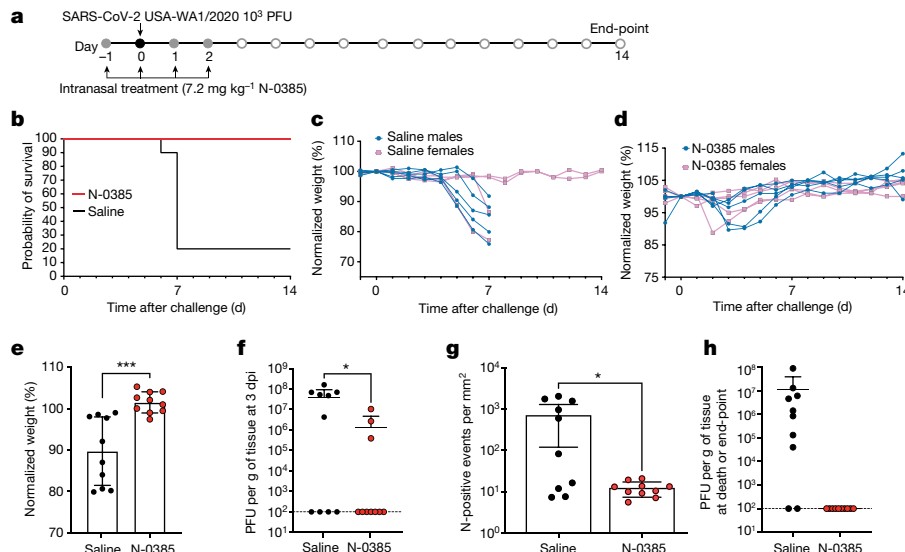

**Fig. 5 | N-0385 reduces viral burden and weight loss and completely prevents mortality in a K18-hACE2 mouse model of SARS-CoV-2 after an early four-day treatment regimen. a**, K18-hACE2 mice were treated once daily on day −1 to day 2 relative to SARS-CoV-2 infection; surviving mice were euthanized on day 14. **b**, Probability of survival. **c**, Weight change of saline control mice. **d**, Weight change of N-0385-treated mice. **e**, Differences in weight loss between treatment groups from **c**, **d** at 6 dpi ($n = 10$ mice (5 males and 5 females) per group); ***$P = 0.0004$. Two-tailed unpaired $t$-test was used to determine significance. Data are mean ± s.d. **f**, Virus titres (PFU per g of tissue) from lungs of infected mice at 3 dpi ($n = 10$ mice per group). Titres were significantly lower for the N-0385 group compared to the saline control group; *$P = 0.0290$. Two-tailed Student's $t$-test was used to determine significance. Data are mean ± s.d. Plaque assays were performed twice per sample from each

mouse and the average was used to determine the PFU per g. **g**, Numbers of events per mm$^2$ that were positive for SARS-CoV-2 nucleocapsid (N) by IHC staining in lung tissue at 3 dpi; the reduction in positive cells was significantly greater for the N-0385 treatment versus the saline control; *$P = 0.0433$. Two-tailed Mann–Whitney test was used to determine significance. Data are mean ± s.d. One complete lung section per mouse was analysed ($n = 10$ mice per treatment group). **h**, Virus titres (PFU per g of tissue) from the lungs of infected mice ($n = 10$ mice per group) at the time of death or the study end-point. Statistical analysis was not performed as samples are from different time points. Data are mean ± s.d. (logarithmic scale precludes negative values being shown). Plaque assays were performed twice per sample from each mouse and the average was used to determine the PFU per g.

A reduction of more than 50% in viral titres and IHC viral staining was observed in N-0385-treated mice compared to the control saline-treated group. Similarly, total pathology scores of lung tissue assessed using IHC sections were improved by approximately 1.9-fold (or 46%) (Fig. 6g, Extended Data Table 5). Together, the in vivo data strongly suggest that N-0385 considerably prevents morbidity and mortality and reduces viral burden in the K18-hACE2 mouse model of severe SARS-CoV-2 infection, when used as a prophylactic or therapeutic treatment.

## Discussion

In this study, we report on N-0385—a potent small-molecule protease inhibitor of human TMPRSS2 and a SARS-CoV-2 pan-variant HDA that is effective in vivo against the B.1.617.2 (Delta) VOC. N-0385 acts as an inhibitor of the TTSP-dependent proteolytic activation of virus spike protein, a critical step in permitting viral–cell membrane fusion and entry into target cells[4]. The nanomolar potency of N-0385 against SARS-CoV-2 infection in human Calu-3 cells and patient-derived colonoids without detectable toxicity yields a selectivity index of greater than 10$^6$. Furthermore, in the K18-hACE2 mouse model, treatment with N-0385 resulted in complete protection against SARS-CoV-2 induced mortality and also provided substantial protection against weight loss, lung pathology and viral infection when treatment occurred at the time of, or 12 h after, infection with B.1.617.2. These data suggest that N-0385 may provide an effective early treatment option against emerging SARS-CoV-2 VOCs.

We have previously shown how peptidomimetic-based compounds with ketobenzothiazole warheads exhibit potent antiviral efficacy in impeding the infection of Calu-3 cells with influenza A H1N1 virus, through inhibition of TTSPs[28]. The activation of the influenza A virus surface glycoprotein hemagglutinin is notably similar to that of the SARS-CoV-2 spike protein, in that both are viral surface protein

homotrimers cleaved by proteolytic enzymes of the TTSP family that are expressed by host epithelial cells[14,40]. TTSPs are attractive broad-spectrum, HDA drug targets because of (i) their central role in mediating viral entry[5]; (ii) their accessibility on the surface of nasal and pulmonary epithelial cells[41,42]; and (iii) their demonstrated therapeutic potential for combating viruses such as SARS-CoV-2 and other human coronaviruses, as well as influenza viruses[14,40,43].

In this work, we present the design and use of peptidomimetics with ketobenzothiazole warheads, which led to the identification of N-0385—a compound with potent inhibitory activity against TMPRSS2 proteolytic activity (IC$_{50}$ = 1.9 nM). When we screened selected TMPRSS2 inhibitors for antiviral activity against SARS-CoV-2, a similar inhibitory profile was observed against TMPRSS2 expressed in Vero E6 cells compared to SARS-CoV-2 infection in Calu-3 cells. N-0385, the lead antiviral candidate, showed potent inhibition of SARS-CoV-2 infection in Calu-3 cells, with an EC$_{50}$ of 2.8 ± 1.4 nM and a selectivity index of higher than $1 \times 10^6$. The potency of N-0385 was validated using two viral biomarkers of intracellular infection as well as by measuring the release of infectious viral particles. Furthermore, complete inhibition of infection was achieved with 100 nM N-0385 in colonoids derived from human donors, confirming the low nanomolar potency of N-0385 against SARS-CoV-2. This complements a recent report that showed that peptidomimetic compounds targeting TMPRSS2 have high potency against SARS-CoV-2-induced cytopathic effects, as well as excellent stability and safety in mice[44].

The usefulness of N-0385 needs to be considered in the context of circulating SARS-CoV-2 variants. VOCs such as B.1.1.7, P.1 and B.1.617.2 were of concern because of their rapid rise to dominance, as well as their extensive spike mutations, which could lead to conformational changes of the trimeric spike structure, which in turn may be detrimental to antiviral effectiveness and vaccine protection[2,11]. In 2021, B.1.617.2 became the dominant circulating variant for which reduced

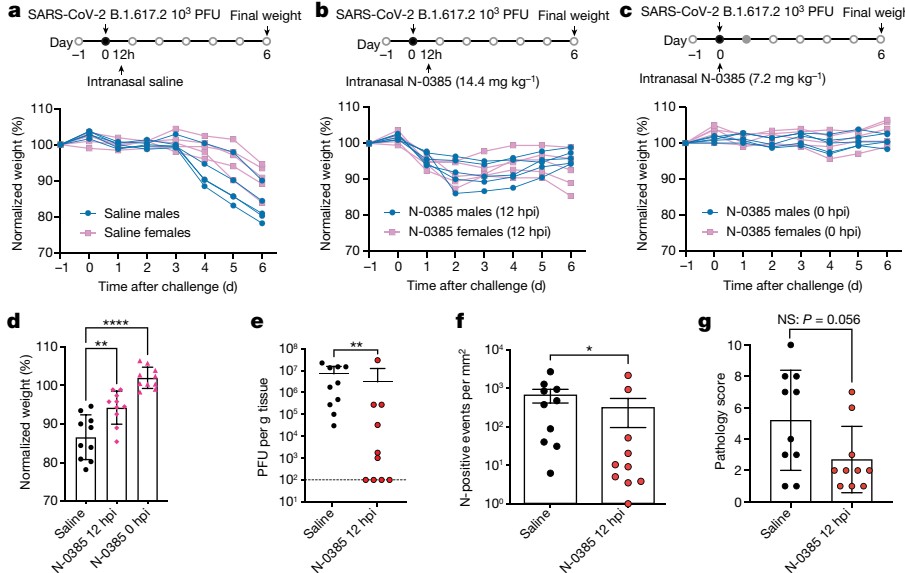

**Fig. 6 | A single dose of N-0385 reduces weight loss and viral burden in K18-hACE2 mice infected with the SARS-CoV-2 B.1.617.2 (Delta) VOC. a,** Weight change in K18-hACE2 mice treated once with saline at 12 hpi. **b,** Weight change in mice treated once with N-0385 at 12 hpi (14.4 mg kg⁻¹). **c,** Weight change in mice treated once with N-0385 at 0 hpi (7.2 mg kg⁻¹). **d,** Differences in weight loss across three treatment groups (**a–c**) at 6 dpi ($n = 10$ mice per treatment group); **P* = 0.0012; ***P* ≤ 0.0001. One-way ANOVA using Dunnett's multiple comparisons test was used to assess significance. Data are mean ± s.d. **e,** Virus titres (PFU per g of tissue) from the lungs of infected mice 3 dpi. Titres were significantly lower for the N-0385 group compared to the saline control group; **P* = 0.0081. Two-tailed Mann Whitney test was used to determine significance. Data are mean ± s.d., $n = 10$ mice per group. Plaque assays were performed twice per sample from each mouse and the average was used to determine the PFU per g. **f,** Numbers of events per mm² that were positive for SARS-CoV-2 nucleocapsid (N) by IHC staining in lung tissue at 3 dpi; the reduction in positive cells was significantly greater for the N-0385 treatment versus the saline control; *$P = 0.0355$. Two-tailed Mann–Whitney test was used to determine significance. Data are mean ± s.d. One whole lung section was analysed per mouse ($n = 10$ mice per group). **g,** Total lung pathology scores of infected mice at 3 dpi as assessed on IHC were improved by approximately 1.9-fold (or 46%) and approach statistical significance; $P = 0.053$. Two-tailed Mann–Whitney test was used to determine significance Data are mean ± s.d. One complete lung section per mouse was analysed ($n = 10$ mice per group).

vaccine efficacy and worse infection outcomes were documented[2]. We hypothesized that the efficacy of N-0385 against four SARS-CoV-2 VOCs (B.1.1.7, P.1, B.1.351 and B.1.617.2) should not be compromised as no mutations in the TMPRSS2 cleavage site have been reported[2,11]. Our results confirmed the low nanomolar pan-variant antiviral activity of N-0385 against these four SARS-CoV-2 VOCs in human cells.

Previous work has shown that the K18-hACE2 mouse model used in our studies is an ideal model for recapitulating the pathology of severe COVID-19 in humans as well as its high morbidity and mortality. SARS-CoV-2 challenge in this model leads to high viral titres in lung and brain tissues with commensurate high morbidity and mortality, weight loss and cytokine and chemokine production[36,45]. Therefore, this model is ideal for testing SARS-CoV-2 therapeutic agents, owing to its severe disease burden as compared to other animal models including mouse-attenuated SARS-CoV-2 in wild-type mice or wild-type SARS-CoV-2 in golden Syrian hamsters, which exhibit milder symptoms. Protection in an animal model with high levels of hACE2, such as the K18-hACE2 mouse model, is thus indicative of the high promise of anti-SARS-CoV-2 antivirals[36]. The mouse TMPRSS2 protein contains 492 amino acids and shares 81.4% similarity and 77.3% identity with human TMPRSS2[46], and we confirmed that our lead peptidomimetic N-0385 inhibited mouse TMPRSS2 with an IC₅₀ of 12.3 ± 1.9 nM.

Intranasal administration has several advantages for the prevention and treatment of SARS-CoV-2 and other viral diseases, including ease of self-administration. SARS-CoV-2 mainly enters the human body through ACE2- and TMPRSS2-positive nasal epithelial cells[47–49]. Intranasal drug delivery maximizes airway and lung exposure while limiting systemic exposure. For example, intranasal administration of a membrane fusion inhibitory lipopeptide prevented the transmission of SARS-CoV-2 in ferrets[50]; however, the efficacy of intranasal delivery of a small molecule inhibitor has not to our knowledge been shown. Under our conditions,

intranasal administration of N-0385 markedly reduced morbidity and mortality in the K18-hACE2 mouse model of severe COVID-19 pathology. We first investigated the survival benefit of an eight-day N-0385 treatment regimen, which protected 70% of the mice from SARS-CoV-2-induced mortality. Once efficacy was established, we investigated a shortened early treatment regimen and observed 100% survival of these mice, which underlines the potent antiviral efficacy of N-0385 and the importance of the TTSP-mediated proteolytic maturation of spike protein for SARS-CoV-2 infection in vivo. In addition to the reduced mortality, morbidity and histological signs, IHC analysis and plaque assays indicated a 98% and 97% reduction in SARS-CoV-2, respectively, in the lungs of N-0385 treated mice at 3 dpi. This is indicative of the effective reduction of virus propagation by N-0385 in this animal model. Although further studies are needed to understand the ideal time points for N-0385 administration, we have shown that N-0385 may also contribute therapeutic efficacy against SARS-CoV-2 VOCs. A single dose at the time of infection or 12 h after infection with B.1.617.2 significantly protected against COVID-19-associated weight loss[36,38], and significantly reduced viral burdens at 3 dpi (by more than 50%), confirming that N-0385 can act as a pan-SARS-CoV-2 prophylactic and therapeutic.

Antiviral candidates for SARS-CoV-2 infection are under investigation in clinical trials and in animal models, but at present, only one study on the DAA GC-376 has reported protection against lethal SARS-CoV-2 infection in the K18-hACE2 model[26]. Plitidepsin, a naturally occurring HDA, protected against lung pathology in the K18-hACE2 model; however, the effect on mortality was not reported[25]. Plitidepsin is a promising HDA that targets the ubiquitously expressed elongation factor 1-alpha 1 and has shown high potency (EC₅₀ = 1.62 nM and selectivity index = 40.4) against SARS-CoV-2 infection in pneumocyte-like cells[25]. Cm and nafamostat mesylate are also HDAs that target serine proteases including host TTSPs and these are also undergoing human trials against SARS-CoV-2; however, no significant protection against infection was observed in the

adenovirus hACE2 model (Cm) of SARS-CoV-2 infection[4,51]. Clinical trial data for the treatment of hospitalized patients with COVID-19 with Cm showed that Cm had no effect on the time to recovery and incidence of death after SARS-CoV-2 infection[19]. Antivirals will probably need to be administered during the very early phase of COVID-19 to be effective in lowering the risk of disease progression, consistent with our short early treatment regimen in K18-hACE2 mice infected with SARS-CoV-2.

Overall, we have developed and characterized N-0385, a highly potent inhibitor of TMPRSS2-like proteases that blocks SARS-CoV-2 VOCs (B.1.1.7, P.1, B.1.351 and B.1.617.2) and is broadly protective against infection and mortality in mice. In addition, we have shown that N-0385 provides an effective early treatment option against SARS-CoV-2 and the B.1.617.2 (Delta) VOC. Moreover, N-0385 analogues may have broader applications in combating other widespread respiratory viruses that usurp TMPRSS2-related proteases for viral entry, including other established coronaviruses, influenza viruses and additional viruses that depend on TTSPs for entering host cells[4,30,40]. We envision a practical use of N-0385 for unvaccinated individuals or those with high risk of exposure or severe disease outcome related to SARS-CoV-2 VOCs and future emerging pathogens. Practically, TTSP inhibitors should be administered as soon as possible after exposure to SARS-CoV-2 for maximal effect and may possibly act synergistically when used in multi-drug combinations with replication inhibitors such as remdesivir, paxlovid and molnupiravir to reduce the risk of antiviral resistance mutations.

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

# Methods

## Cell lines, antibodies and inhibitors

Calu-3 cells[52] (ATCC HTB-55) were cultured according to ATCC recommendations. All experiments were performed in these cells below passage 15. Vero E6 cells (ATCC CRL-1586; used for SARS-CoV-2 plaque assay) were grown in minimum essential medium (MEM) supplemented with 10% FBS, 1 mM sodium pyruvate and 0.1 nM non-essential amino acids, and used at passage <40. All cells were expanded in a T175 flask with 5% carbon dioxide at 37 °C. Cell density was kept between 0.25 and 2 million cells per ml. Vero E6 cells (used for TMPRSS2 proteolytic activity screening and $IC_{50}$ determination) were maintained in Eagle's minimum essential medium (EMEM) containing 10% foetal bovine serum, 2 mmol l[−1] L-glutamine, 100 IU ml[−1] penicillin and 100 μg ml[−1] streptomycin. Cm was obtained from MilliporeSigma. The SARS-CoV-2 nucleocapsid antibody (HL344) (GTX635679) was provided by Genetex; mouse anti-dsRNA antibody (J2-1904) was purchased from Scions English and Scientific Consulting[33]; Hoechst 33342 and secondary antibodies goat anti-mouse IgG Alexa Fluor 488 (A11001) and goat anti-rabbit IgG Alexa Fluor 555 (A21428) were obtained from Invitrogen. Cell lines were screened for mycoplasma contamination using MycoAlert Mycoplasma Detection Kit (Lonza).

## Synthesis of peptidomimetic compounds

Preparation of the compounds using a mixed approach of solution and solid-phase synthesis is described in the Supplementary Information, in addition to a synthetic scheme of analogues, nuclear magnetic resonance (NMR), high-resolution mass spectrometry (HRMS), ultraperformance liquid chromatography–mass spectrometry (UPLC–MS) retention time, structure, purity, and molecular formula strings of compounds. Amino acids and coupling reagents were obtained from Chem-Impex International and used as received. All other reagents and solvents were purchased from Sigma-Aldrich or Thermo Fisher Scientific. Tetrahydrofuran (THF) was dried over sodium benzophenone ketyl; dichloromethane over $P_2O_5$; methanol over magnesium. Celite (AW Standard Super-Cel NF) was obtained from Sigma-Aldrich. Thin-layer chromatography was performed on glass plates covered with silica gel (250 μm) 60 F-254 (Silicycle). Flash chromatography was performed with Silicaflash P60 (40–63 μm, Silicyle). Chlorotrityl chloride (CTC) resin was obtained from Matrix Innovation and generally used with a loading of 1.2 mmol g[−1]. Reactions on resin were conducted in 60 ml polypropylene cartridges (obtained from Applied Separations) and Teflon stopcocks. Reactors were gently rocked on an orbital shaker at 172 rpm during solid-phase chemistry. The resin was washed with the indicated solvent for 2–5 min with 10 ml solvent per gram of resin. Purity was analysed on a Waters UPLC H-Class with UV detection PDA equipped with an Acquity UPLC CSH C18 1.7 μm 2.1 × 50 mm² column. MS spectra were recorded on a Waters SQD 2 detector (electrospray) instrument with a linear gradient of 5–95% $CH_3CN$ and $H_2O$ containing 0.1% formic acid. Final products were purified to higher than 95% purity (UPLC-UV) using a Waters Preparative LC (Sample Manager 2767 (fraction collector); binary gradient module 2545, with two 515 HPLC pumps and a system fluidics organizer (SFO); photodiode array detector 2998: column X Select CSH Prep C18 5 μm OBD 19 × 250 mm² column; buffer: A: 0.1% HCOOH in $H_2O$; B: 0.1% HCOOH in ACN; flow 20 ml min[−1]). The gradient was 10–60% of acetonitrile at a flow rate of 20 ml min[−1]. Purities of all compounds in this paper were higher than 95% as assessed by UPLC.

## Molecular modelling

A homology model of the TMPRSS2 catalytic domain was built using the structure of matriptase (PDB: 6N4T) with the 'Homology Model' module of the Molecular Operating Environment (MOE) from the Chemical Computing Group. Sequence alignment of catalytic domains of matriptase with TMPRSS2 using 'Align Sequences Protein BLAST' and MOE sequence alignment allowed building of a high-quality model. Ten models were created, and the final model was selected using the best score obtained by the generalized-born volume integral/weighted surface area (GBVI/WSA) scoring method[53]. The final model was refined and minimized using the Amber10:Extended Huckel Theory (EHT) force field. After drawing the structure, all protein–ligand complexes were prepared using the Protonate 3D tool; then the partial charges were calculated, and the ligands were energy-minimized.

Molecules were docked in the protein-binding site with the software MOE2019.01.02. All atoms were fixed, and the ligands were allowed to be flexible. The carbon of the ketone making the reversible covalent bond with the protein was fixed at 3.0 ± 0.1 Å of the catalytic serine to constrain the position of the ketobenzothiazole group within the binding site. The guanidine of the arginine in P1 was also fixed through two key interactions in the binding site. Conformational search using LowModeMD was made with AMBER10:EHT as a molecular mechanics force field with default parameters (rejection limit: 100; RMS gradient: 0.05; conformation limit: 10,000; iteration limit: 10,000). Finally, a second round of energy minimization was performed around the ligand-binding site. The low energy conformations of the inhibitor-protein complexes were analysed for their binding interactions.

## TMPRSS2 pericellular activity screening assay and determination of $IC_{50}$

Vero E6 cells were transfected with mock (pcDNA3.1), TMPRSS2 (pcDNA3.1/TMPRSS2 Uniprot: O15393-1) or TMPRSS2(S441A) (pcDNA3.1/TMPRSS2-S441A) using Lipofectamine 3000 (Thermo Fisher Scientific) in 12-well plates. For the mouse TMPRSS2 assay, empty vector (pCMV6-Entry, Origene PS100001) and TMPRSS2-Myc-DDK (Origene MR207852) were used. After 24 h transfection, cells were washed with PBS and the medium replaced with HCell-100 medium (Wisent) containing 200 μM Boc-QAR-AMC (R&D Systems) and either vehicle (0.01% DMSO) or compounds at the indicated concentration for 24 h. To measure proteolytic activity, 90 μl of cell medium was transferred to a black 96-well plate, and fluorescence was measured at room temperature (excitation: 360 nm, emission: 460 nm) using an FLx800 TBE microplate reader (Bio-Tek Instruments). Background-subtracted (mock-transfected cells) proteolytic activities are presented as the percentage of activity relative to vehicle-treated cells (screen at 10 nM). $IC_{50}$ values were determined after generating a nonlinear regression analysis from a log([Compound]) versus a proteolytic activity plot using GraphPad Prism software (v.9.0.1). GraphPad Prism was used to identify and eliminate outliers ($Q = 1$) and assess the goodness of the fits. $IC_{50}$ values presented are the mean ± s.d. of at least three independent experiments.

## SARS-CoV-2 infection and treatment in Calu-3 lung epithelial cells

All infections were carried out in a Biosafety Level 3 (BSL3) facility (UBC FINDER) in accordance with the Public Health Agency of Canada and UBC FINDER regulations (UBC BSL3 Permit B20-0105 to F.J.). SARS-CoV-2 (SARS-COV-2/Canada/VIDO-01/2020; VIDO) was provided by S. Mubareka. SARS-CoV-2 VOCs (B.1.1.7, B.1.351, P.1 and B.1.617.2) were provided by M. Krajden. SARS-CoV-2 VOCs were first isolated in Vero-TMPRSS2 cells (passage 1) and then passaged in Vero E6 cells (passage 2). Viral stocks used in the experiments (passage 3) were propagated in Vero E6 cells[54]. For experiments, passage three of the virus was used with a determined viral titre of $1.5 × 10^7$ PFU ml[−1]. Calu-3 cells were seeded at a concentration of 10,000 cells per well in 96-well plates the day before infection. SARS-CoV-2 stocks were diluted in cell-specific medium to a multiplicity of infection (MOI) of 2. Cells were pretreated with compounds for 3 h and then incubated with the virus for 2 days, followed by fixation of the cells with 3.7% formalin for 30 min to inactivate the virus. The fixative was removed, and cells

were washed with PBS, permeabilized with 0.1% Triton X-100 for 5 min, and blocked with 1% bovine serum albumin (BSA) for 1 h, followed by immunostaining with the mouse primary antibody J2 (dsRNA) and rabbit primary antibody HL344 (SARS-CoV-2 nucleocapsid) at working dilutions of 1:1,000 for 1 h at room temperature. Secondary antibodies were used at a 1:2,000 dilution and included the goat anti-mouse IgG Alexa Fluor 488 and goat anti-rabbit IgG Alexa Fluor 555 with the nuclear stain Hoechst 33342 at 1 μg ml$^{-1}$ and F-actin staining with Alexa Fluor 647 phalloidin at a 1:300 dilution for 1 h at room temperature in the dark. After washing with PBS, plates were kept in the dark at 4 °C until imaging on a high-content screening (HCS) platform (CellInsight CX7 HCS, Thermo Fisher Scientific) with a 10× objective, or an EVOS M7000 Imaging System (Thermo Fisher Scientific) with a 20× or 40× objective. Confocal imaging was performed with a Leica TCS SP8 STED 3× laser scanning confocal microscope (Leica) equipped with a 100×/1.4 Oil HC PL APO CS2 STED White objective, 405-nm laser, a white light laser and HyD detectors, and operated with Leica Application Suite X (LAS X) software. Three-dimensional (3D) volume rendering was done with LAS-X. Two-dimensional images were exported into tiff format. Merging of different channels and the addition of the scale bar were performed using ImageJ/Fiji.

## High-content screening of SARS-CoV-2 infection
Monitoring of the total number of cells (based on nuclei staining) and the number of virus-infected cells (based on dsRNA and nucleocapsid staining) was performed using the CellInsight CX7 HCS platform (Thermo Fisher Scientific), as previously described[55,56]. In brief, nuclei are identified and counted using the 350/461 nm wavelength (Hoechst 33342); cell debris and other particles are removed on the basis of a size filter tool. A region of interest (ROI, or 'circle') is then drawn around each host cell and validated against the bright field image to correspond with host cell membranes. The ROI encompasses the 'spots' where dsRNA (485/521 nm wavelength) and SARS-CoV-2 nucleocapsid (549/600 nm wavelength) are localized. Finally, the software (HCS Studio Cell Analysis Software, v.4.0) identifies, counts and measures the pixel area and intensity of the 'spots' within the 'circle'. The fluorescence measured within each cell (circle) is then added and quantified for each well. The total circle spot intensity of each well corresponds to intracellular virus levels ($Z' > 0.7$) and is normalized to non-infected cells and to infected cells with 0.1% DMSO. Nine fields were sampled from each well. Nuclei stain (Hoechst 33342) was also used to quantify cell loss (owing to cytotoxicity or loss of adherence) and to verify that the changes in viral infection did not result from a decrease in cell numbers.

## Median EC$_{50}$ curves
Intracellular dose–response (EC$_{50}$ values) for selected compounds against SARS-CoV-2 were determined by pretreating Calu-3 cells for 3 h with serially diluted compounds (0.064, 0.32, 1.6, 8, 40, 200 and 1,000 nM), followed by SARS-CoV-2 infection for 48 h. Viral infection was detected by staining for dsRNA or nucleocapsid signal and quantified as described above. EC$_{50}$ experiments were repeated at least three times for each compound with three technical replicates in each experiment. Intracellular nucleocapsid levels were interpolated to negative control (0.1% DMSO, no infection) = 0 and positive control (0.1% DMSO, with infection) = 100. The GraphPad Prism 9 (GraphPad Software) nonlinear regression fit modelling variable slope was used to generate a dose–response curve ($Y = \text{Bottom} + (\text{Top} - \text{Bottom})/(1 + 10^{(\log IC50 - X) \times \text{HillSlope}})$, constrained to Top = 100, Bottom = 0.

## SARS-CoV-2 plaque assay
A total of 250,000 Vero E6 cells were seeded in complete MEM medium in 6-well plates and incubated for 24 h at 37 °C before infection with a 1:1,000 dilution of supernatant from mock, infected, and treated and infected cells. The wells were washed once with PBS before 100 μl virus dilution was added per well in quadruplicate. Infected cells were

incubated at 37 °C for 1 h, mixed gently every 15 min, then covered with 2 ml overlay medium of 2% Avicel CL-611 (DuPont Pharma Solutions) or 0.6% agar diluted 1:1 with 2× MEM (Gibco). The cells were then incubated for 3 days. To fix the cells, 2 ml 8% formalin or 4% PFA was added to each well for 30 min, following removal of the overlay/formalin solution. Cells were gently washed with 1 ml tap water per well, followed by staining with 200 μl 1% crystal violet in 20% methanol for 5 min or in 0.5% crystal violet in 30% methanol for 15 min. Crystal violet was removed, and the cells were washed three times with 1 ml tap water per well, then dried before the viral plaques were manually counted.

## Cytotoxicity assays
Calu-3 and Vero E6 cells (2500 or 10,000 cells for samples, 80–20,000 cells for standard curve) were seeded in 96-well plates. Following a 24-h incubation at 37 °C, 5% CO$_2$, cells were washed with D-PBS and compounds added (10 μM) for an additional 24-h incubation. Cellular viability was assessed using Cell Titer-Glo 2.0 Cell Viability Assay (Promega) according to the manufacturer's instructions. The number of viable cells was extrapolated using the standard curve. Cellular viability in Vero E6 cells was expressed relative (%) to vehicle-treated cells. Data are from four independent experiments (mean ± s.d.).

## Protease selectivity of N-0385
Recombinant human matriptase, hepsin and DESC1 were expressed and purified as described previously[57,58]. Recombinant human furin, human cathepsin L (Bio-Techne), and human thrombin (MilliporeSigma) were obtained from commercial sources. $K_i$ values were determined using steady-state velocities as previously reported[27,29]. Assays were performed at room temperature in assay buffers (50 mM Tris-HCl pH 7.4; 150 mM NaCl; 500 μg ml$^{-1}$ BSA for matriptase, hepsin, DESC1 and thrombin; 50 mM HEPES pH 7.4, 1 mM β-mercaptoethanol, 1 mM CaCl$_2$, 500 μg ml$^{-1}$ BSA for furin; 50 mM MES pH 6, 5 mM DTT, 1 mM EDTA, 0.005% Brij 35, 500 μg ml$^{-1}$ BSA for cathepsin L). To measure proteolytic activity, protease (0.25 to 1 nM) was added to the assay buffer containing different concentrations of compounds and a fluorogenic substrate (Boc-RVRR-AMC for furin, Z-LR-AMC for cathepsin L, and Boc-QAR-AMC for the other proteases). Activity was monitored (excitation: 360 nm; emission: 460 nm) using a FLx800 TBE microplate reader (Bio-Tek Instruments). If substantial inhibition occurred using a ratio $I/E \leq 10$ plots of enzyme velocity as a function of the inhibitor, concentrations were fitted by nonlinear regression analysis to the Morrison equation for tight-binding inhibitors. If inhibition occurred only at $I/E > 10$, plots of enzyme velocity as a function of substrate concentration at several inhibitor concentrations were fitted by nonlinear regression to equations describing different models of reversible inhibition (competitive, uncompetitive, non-competitive and mixed model). The preferred model was used for $K_i$ determination. $K_i$ was calculated from at least three independent experiments (mean ± s.d.). The maximum concentration of compounds used for the assays was 10 μM.

## SARS-CoV-2 infection of human biopsy-derived colonoid monolayers
Intestinal biopsy-derived colonoids from healthy donors were obtained from the Johns Hopkins Conte Digestive Disease Basic and Translational Research Core Center (NIH NIDDK P30-DK089502) and grown as described previously[34]. In brief, human colonoid monolayers were generated by combining the colonoids from one Matrigel dome (around 100 or more colonoids in a 25-μl dome). Domes were dislodged with a cell scraper in 1 ml of Cultrex Organoid Harvesting solution (Bio-techne, R&D Systems brand, 3700-100-01) and incubated for 1 h at 4 °C on a shaker at 250 rpm. After incubation, cells were diluted with an equal volume of complete medium without growth factors (CMGF; Advanced DMEM/F-12 (Gibco brand, Thermo Fisher Scientific 123634010), 10 mM HEPES (Invitrogen 15630-080), GlutaMAX (Gibco brand, 35050-061), and 100 U ml$^{-1}$ of penicillin–streptomycin (Gibco brand, 15140-122)),

and then centrifuged at 400$g$ for 10 min at 4 °C. Cells were resuspended in 50 µl per well of TrypLE Express (Invitrogen, 12604021) and then incubated for 1 min at 37 °C. After incubation, 10 ml of cold CMGF was added and the cells were pelleted by centrifugation as above and then resuspended in 100 µl per well of monolayer medium (IntestiCult Organoid Growth Medium (Human) 06010), 10 µM of Rho Kinase inhibitor, Y-27632 (Stemcell 72304) and 50 µg ml$^{-1}$ of gentamicin (Gibco brand, Thermo Fisher Scientific, 1510064)). They were then seeded at a 1:4 dome-to-well ratio in a 96-well plate coated in 100 µl of 34 µg ml$^{-1}$ human collagen IV (Sigma C5533). Cells were fed every two days and were used for experiments after they were fully confluent (four to five days). Cells were treated with compounds for 3 h before SARS-CoV-2 infection (SARS-COV-2/Canada/VIDO-01/2020; MOI ≈ 1) for 72 h, and then were fixed and stained for nucleocapsid and dsRNA as described in 'SARS-CoV-2 infection and treatment in Calu-3 lung epithelial cells'. Quantification of infected cells was performed as described above in 'High-content screening of SARS-CoV-2 infection'. Imaging was performed on the EVOS M7000 microscope using the following channels: 357/447 nm for nuclear staining (Hoechst 33342), 470/525 nm for dsRNA (Alexa Fluor 488), and 531/593 nm for nucleocapsid (Alexa Fluor 555).

## PCR
Total RNA extractions in Calu-3 cells were performed using TRIzol (Invitrogen) with chloroform as recommended by the manufacturer's protocol. The aqueous layer was recovered, mixed with 1 volume of 70% ethanol, and applied directly to an RNeasy Mini kit column (Qiagen). RNA quality and the presence of contaminating genomic DNA were verified as described previously[59]. RNA integrity was assessed with an Agilent 2100 Bioanalyzer (Agilent Technologies). Reverse transcription was performed on 1.1 µg of total RNA with Transcriptor reverse transcriptase, random hexamers, dNTPs (Roche Diagnostics) and 10 units of RNaseOUT (Invitrogen) following the manufacturer's protocol in a total volume of 10 µl. For colonoids, cells were rinsed in ice-cold PBS and lysed using the miRNeasy Mini Kit (Qiagen) per the manufacturer's instructions. RNA was isolated and reverse-transcribed using Quanta Biosciences qScript cDNA SuperMix.

For the qPCR assays, primers were individually resuspended to 20–100 µM stock solutions in Tris-EDTA buffer and diluted as a primer pair to 1 µM in RNase DNase-free water. qPCR was performed in 10 µl in 96-well plates on a CFX OPUS-96 thermocycler (Bio-Rad) with 5 µl of 2× PerfeCTa SYBR Green Supermix (Quantabio), 10 ng (3 µl) of cDNA and 200 nM (final concentration; 2 µl) primer pair solutions. The following cycling conditions were used: 3 min at 95 °C and 50 cycles of 15 s at 95 °C, 30 s at 60 °C and 30 s at 72 °C. Relative expression levels were calculated according to the qBASE framework[60] with YWHAZ, PUM1 and MRPL19 as housekeeping genes for normalization. Primer design and validation were evaluated as described elsewhere[59]. In every qPCR run, a no-template control was performed for each primer pair; these were consistently negative. All qPCR assays were performed by the RNomics Platform of the Université de Sherbrooke.

## SARS-CoV-2 infection and treatment in mice
Animal studies were performed in accordance with the recommendations in the Guide for the Care and Use of Laboratory Animals of the National Institutes of Health. All protocols were performed under approved BSL-3 conditions and approved by the Institutional Animal Care and Use Committee at Cornell University (IACUC mouse protocol 2017-0108 and BSL3 IBC MUA-16371-1). Intranasal virus and antiviral treatments were performed under anaesthesia, and all efforts were made to minimize animal suffering. Eight-week-old heterozygous K18-hACE2 c57BL/6J mice (strain: 2B6.Cg-Tg(K18-hACE2)2Prl mn/J)[37,61,62] were used for this study (Jackson Laboratory). Mice were intranasally inoculated with $1 \times 10^3$ PFU per mouse using passage 1 of single-plaque isolated virus propagated in Vero-TMPRSS2 cells from isolate USA-WA1/2020 (BEI resources; NR-52281) or isolate NYI31-21

(B1.617.2 Delta VOC) isolated by the laboratory of D.G.D. Mice were housed in groups of five per cage and fed a standard chow diet. Daily treatments were administered intranasally at 7.2 mg kg$^{-1}$ or 14.4 mg kg$^{-1}$ using the average weights of each group separated by sex. Mice were monitored and weighed daily and humanely euthanized at predetermined criteria to minimize distress after approved protocols, generally when weight loss reached 20% from day of challenge or mice became moribund with a clinical score of greater than 3 on a 5-point scale[63]. Mouse lungs and brains were collected directly after euthanasia and placed in DMEM with 2% FBS for plaque assays[63]. For each experiment, 10 mice (5 males and 5 females) were used per treatment group, for determination of statistical significance and gender trends. Mice were designated into groups randomly to reduce bias due to differences in weight, and animal studies were performed using age-matched mice to compare across groups. Investigators were not blinded to groups during weighing and scoring of animal health owing to the nature of the infectious agent.

## Mice histopathology
For histological examination, mouse lungs and brains were collected directly after euthanasia and placed in 10% formalin for more than 72 h, after which tissues were embedded in paraffin. Tissue sections (4 µm) were analysed after staining with H&E and scored blinded by an anatomic pathologist. For lung, scores were applied based on the percentage of each tissue type (alveolus, vessels and so on) affected using the following criteria: (0) normal; (1) less than 10% affected; (2) 10–25% affected; (3) 26–50% affected; and (4) more than 50% affected[64]. Assessment of lung pathology in K18-hACE2 mice infected with the SARS-CoV-2 B.1.617.2 (Delta) VOC after single-dose treatment was performed on IHC sections counterstained with haematoxylin. For brains, histological scoring was assessed for perivascular inflammation using the most severely affected vessel and the following criteria: (0) no perivascular inflammation; (1) incomplete cuff one cell layer thick; (2) complete cuff one cell layer thick; (3) complete cuff two to three cells thick; and (4) complete cuff four or more cells thick. Necrotic cells in the neuroparenchyma were assessed per 0.237 mm$^2$ field using the most severely affected area and the following criteria: (0) no necrotic cells; (1) rare individual necrotic cells; (2) fewer than 10 necrotic cells; (3) 11 to 25 necrotic cells; (4) 26 to 50 necrotic cells; and (5) greater than 50 cells.

To detect viral antigen, sections were labelled with anti-SARS-CoV-2 nucleocapsid protein rabbit IgG monoclonal antibody (GeneTex; GTX635679) at a 1:5,000 dilution and processed using a Leica Bond Max automated IHC stainer. Leica Bond Polymer Refine Detection (Leica; DS9800) with DAB was used as the chromogen. Image acquisition was performed using a Roche Ventana DP200 slide scanner. Digital image analysis was performed using QuPath software[65,66] v.0.2.3. Tissues were annotated to include all available lung tissue or all brain tissue excluding cerebellum, as cerebellar tissue was not available for all mice. Following annotation, automated detection was performed using automated SLIC superpixel segmentation with a DAB mean detection threshold of 0.18892.

For each experiment, 10 mice (5 males and 5 females) were analysed per treatment group, for determination of statistical significance and gender trends. Representative images were selected on the basis of the prevalent trend for a given treatment group, showing images representative of the mean pathological score.

## Reporting summary
Further information on research design is available in the Nature Research Reporting Summary linked to this paper.

## Data availability
Correspondence and requests for material related to cell-based SARS-CoV-2 studies should be addressed to F.J., those related to

peptidomimetics and in vitro studies should be addressed to R.L. and those related to animal studies should be addressed to H.C.A. Source data are provided with this paper.

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

**Acknowledgements** We acknowledge the support of the CL-3 facility (Facility for Infectious Disease and Epidemic Research (FINDER) of the Life Sciences Institute of the University of British Columbia, which was founded by F.J.; and its biosafety support staff, including B. Ahidjo and T. Dean Airey. We thank the LSI imaging facility of the Life Sciences Institute of the University of British Columbia, funded by the Canadian Foundation of Innovation and BC Knowledge Development Fund, as well as a Strategic Investment Fund (Faculty of Medicine, University of British Columbia). We also thank A. Ball for supplying the SARS-CoV-2 (COVID-19) nucleocapsid antibody (HL344) (GTX635679); S. Mubareka for providing the original SARS-CoV-2 strain (SARS-COV-2/Canada/VIDO-01/2020); and M. Petric, S. Kaweski, P. Levett and M. Krajden for isolating and providing the SARS-CoV-2 VOCs (B.1.1.7, B.1.351, P.1 and B.1.617.2). We acknowledge a donation towards the purchase of the CellInsight CX7 HCS system provided by the Vancouver General Hospital Foundation to F.J. We thank N. Durso for CX7 expertise and support; the Cornell BSL-3, animal and biosafety support staff and members of the laboratory of A.A. for their support in setting up the animal studies, including but not limited to A. L. Redko, N. Kushner, P. Jennette, J. Turse, B. Singh, D. Miller, D. G. Collins and T. L. Van Deusen; NEOMED/adMare in the early phases of developing host-based antivirals; E. Tan for help growing and preparing cells for plaque assays; E. Colombo and B. Plancq for the synthesis of some intermediates, analogues and warhead; J. Kelly for proofreading the manuscript; N. Zachos from John Hopkins University for providing human colonoids; and the patients who donated these samples for research. We recognize the help of the RNomics Platform of the Université de Sherbrooke. We dedicate this work to the memory of Professor Eric Marsault[67]. This work was supported by operating grants from the Canadian 2019 Novel Coronavirus (COVID-19) Rapid Research Funding program of the Canadian Institutes of Health Research (CIHR) (UBR 322812; VR3-172639 (R.L., P.L.B. and F.J.) and OV3-170342 (F.J.)); by a Genome British Columbia/COVID-19 Rapid Response Funding Initiative (COVO11 (F.J.)); by the Coronavirus Variants Rapid Response Network (CoVaRR-Net) (175622 (F.J. and A.D.O.)); by a Cornell University Seed Grant, Cornell University start-up funds and a George Mason University, Mercatus Center; Emergent Ventures—Fast Grant (H.A.C.); by a National Institutes of Health research grant (R01AI35270 (G.W.)); by a MITACS Accelerate Fellowship (IT18555 (W.R.)); an NIH training grant (T32EB023860 (A.A. to support D.W.B.) and R25GM125597 (A.A. to support B.I.)); by an NIH R01 (AI138570 (A.A.)); by a CIHR Frederick Banting and Charles Best Canada Graduate Scholarship Award (167018 (S.P.D.)); by a PROTEO graduate scholarship (T.V.); and by a MITACS Inc. Accelerate fellowship COVID-19 Award (IT18585 (T.S.)).

**Author contributions** S.P.D., D.W.B., B.I. and A.D.O. contributed equally to this work. R.L. and F.J. conceptualized the initiation of the study. F.J. and T.S. conceptualized the cell-culture based SARS-CoV-2 infection experiments. T.S. performed all cell-based SARS-CoV-2 infection experiments, including plating, treating, fixing, staining and scanning cells. T.S. prepared viral stocks of SARS-CoV-2 and VOCs used in cell-based experiments, performed plaque assays and quantified viral titres. T.S. performed all the antiviral screening and ED₅₀ determination experiments in Calu-3 cells. C.A.H.T. also plated cells (for dose–response curves, confocal imaging, viral propagation and plaque assays), treated cells with compounds, stained cells for CX7 and confocal microscopy and scanned plates on CellInsight CX7. W.D.R. prepared colonoids for SARS-CoV-2 infection. H.C.A. and I.A.M. conceptualized the animal experiments. I.A.M. prepared the WA1/2020 virus used in plaque assays and animal experiments, and performed the animal experiments, collected samples for and generated the data in Figs. 4, 5, 6e–g, Extended Data Fig. 5. M.J. performed all the histopathology and IHC analysis. D.W.B. and B.I. performed the plaque assays for Fig. 5f, h, Extended Data Fig. 5c, and the animal experiments for Fig. 6a–d. M.M. and D.G.D. isolated and prepared the virus stock of SARS-CoV-2 B.1.617.2 for use in the animal experiments in Fig. 6. J.S. and A.A. maintained and provided hACE2-K18 mice for the studies used in Fig. 5. T.V. and P.-L.B. performed the synthesis of the peptidomimetic compounds. P.-L.B. designed the peptidomimetic compounds, coordinated the synthesis and structure–activity relationship studies and performed molecular modelling. S.P.D. performed all the inhibitor screening and IC₅₀ determination experiments in VeroE6 cells. A.D. performed viability assays in Vero E6 cells and in vitro selectivity of N-0385 (Kᵢ). Visualizations and figures were made by T.S., A.D.O., S.P.D., A.D., G.G., A.C., P.-L.B., I.A.M. and M.J. Funding was obtained by R.L., F.J., H.C.A., G.R.W., A.A., E.M., I.R.N. and A.D.O. A.D., A.D.O. and P.-L.B. contributed to project administration. A.D.O. coordinated writing, assembly, submission and revisions of the manuscript. The research was performed under the supervision of F.J., R.L., H.C.A., P.-L.B., G.R.W., E.M., A.A., G.R.V.d.W., D.G.D. and I.R.N. The original manuscript was written by A.D.O., F.J., P.-L.B., R.L., A.D., H.C.A., I.A.M. and M.J. Editing and review of the manuscript was performed by T.S., I.A.M., B.I., S.P.D., G.R.W., R.L., P.-L.B., A.D., H.C.A., A.C., I.A.M., G.R.V.d.W., A.A., M.J., I.R.N., A.D.O. and F.J.

**Competing interests** P.-L.B. and R.L. are inventors on patent applications (US9365853B2 and US10988505B2) that cover matriptase and other type II transmembrane serine proteases inhibitors for treating and preventing viral infections, respiratory disorders, inflammatory disorders, pain disorders, tissue disorders, hyperproliferative disorders, and disorders associated with iron overload. The remaining authors declare that they have no competing interests.

**Additional information**
**Correspondence and requests for materials** should be addressed to Richard Leduc, Hector C. Aguilar or François Jean.

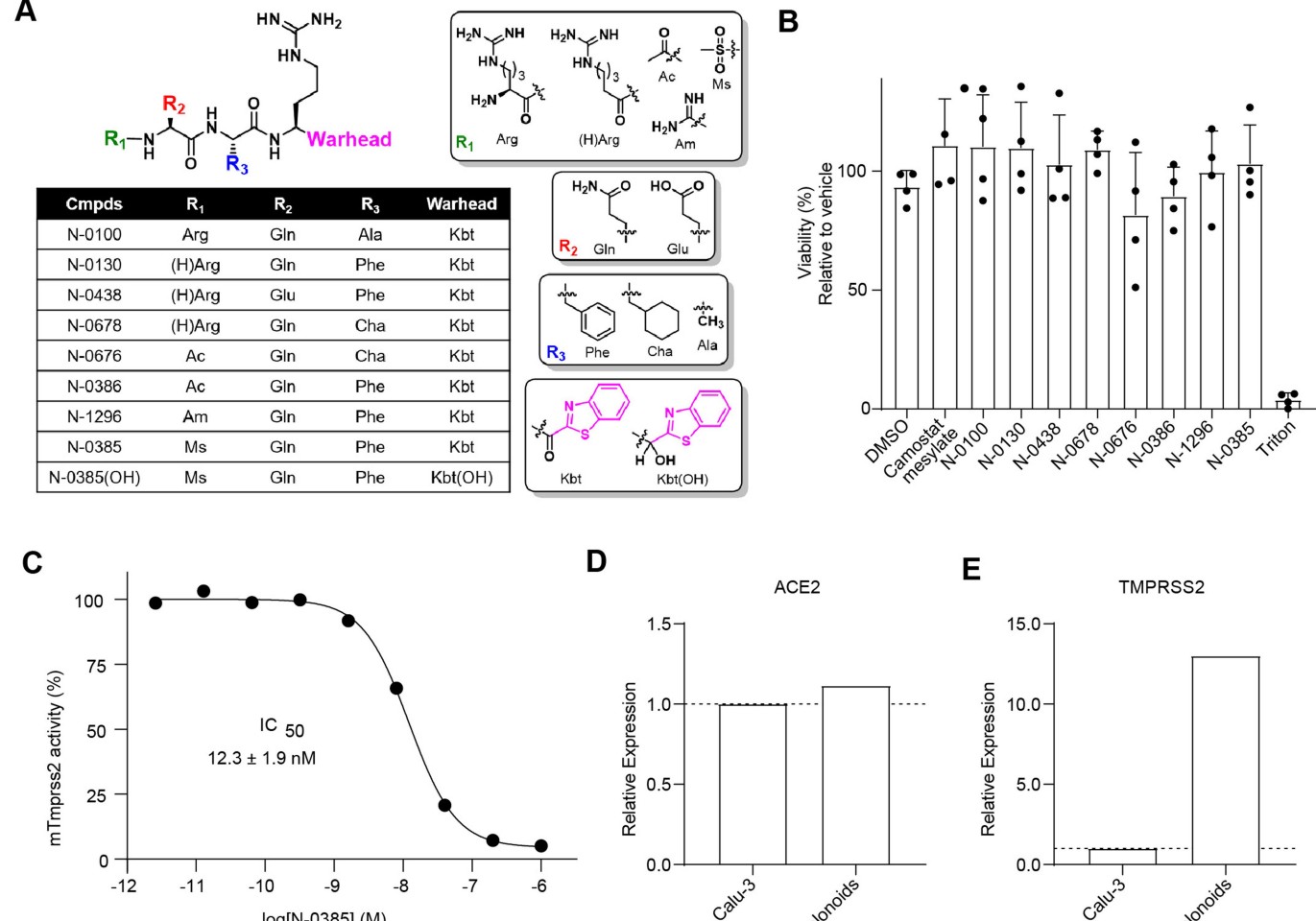

**Extended Data Fig. 1 | Characterization of peptidomimetic inhibitors. (A)** Backbone structure of peptidomimetic compounds used in this study along with the groups used in the N-terminal R1 position, R2, R3 and the C-terminal warhead (ketobenzothiazole; kbt) and alcohol warhead (Kbt(OH)). The P1 position is Arginine in all compounds. (**B**) Cytotoxicity of compounds in Vero E6 cells. Cellular viability was evaluated after 24 h exposure to 10 μM of the indicated compounds (n = 4 independent experiments performed in duplicate). Results are background corrected and presented as the mean viability (%) ± standard deviation (SD) compared to one replicate of vehicle treated cells (DMSO 0.01%) in each experiment. Triton X-100 0.01% was used as a toxicity control. (**C**) N-0385 inhibition of mouse TMPRSS2 (mTMPRSS2). Vero E6 cells were transfected with either an empty vector or mTMPRSS2 for 24 h.

N-0385 was added concomitantly with a fluorogenic substrate on cells for an additional 24 h before fluorescence reading. Relative activity was measured using the mock-subtracted fluorescence and reported as the percentage of residual activity. Dose–response curves were generated and $IC_{50}$ values were determined using nonlinear regression analysis. One representative $IC_{50}$ curve is shown. The $IC_{50}$ value shown represents the mean ± SD from n = 3 independent experiments. (**D**, **E**) Real-time PCR analysis for relative expression of (**D**) ACE2 and (**E**) TMPRSS2 expression in Calu-3 cells and colonoids. Relative expression levels normalized using 3 housekeeping genes (YWHAZ, PUM1, MRPL19) are shown for one sample (RNA extract) performed in technical triplicates.

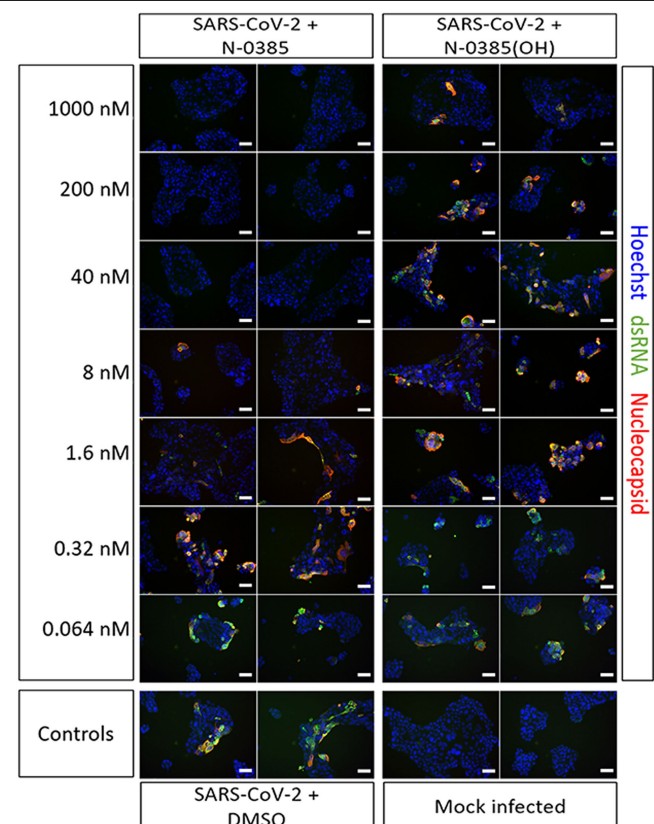

**Extended Data Fig. 2 | N-0385 inhibits SARS-CoV-2 infection in a dose-dependent manner.** Representative images (CellInsight CX7 High Content Screening platform) are shown from Calu-3 cells treated with the indicated doses of N-0385 and N-0385(OH) for 3 h prior to infection with SARS-CoV-2 (VIDO) for two days. Cells are stained for Hoechst 33342 (blue), dsRNA (green), nucleocapsid (red). Each image represents one of nine fields of view from a single well of a 96-well plate. Each independent experiment was performed n = 8 for N-0385 and n = 5 N-0385(OH), with 3 wells of each condition analysed per experiment. Scale bars = 50 μm.

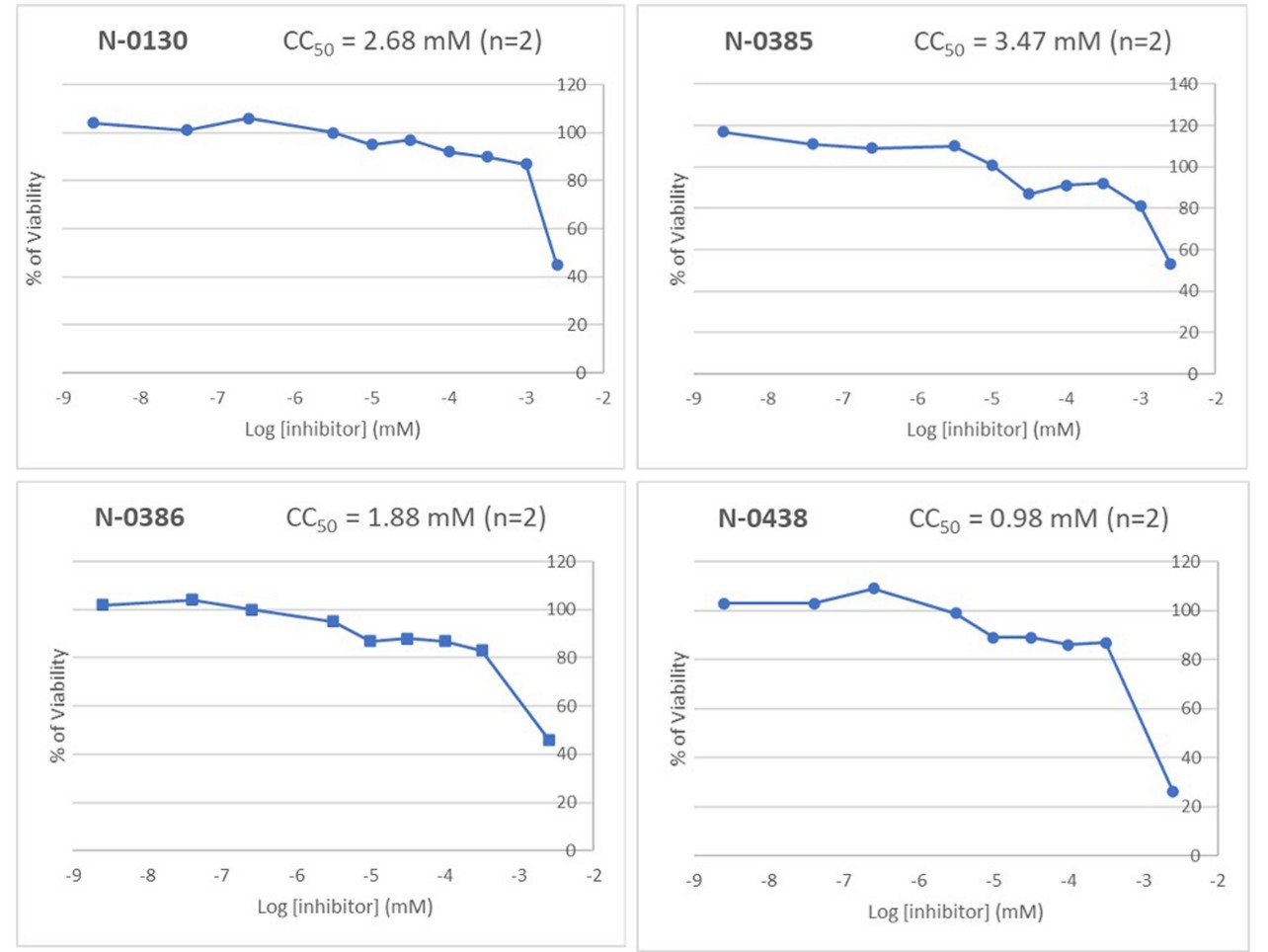

**Extended Data Fig. 3 | Dose–response CC_50 curves for N-130, N-0385, N-0386 and N-0438 in Calu-3 lung epithelial cells.** n = 2 independent experiments were performed for each compound.

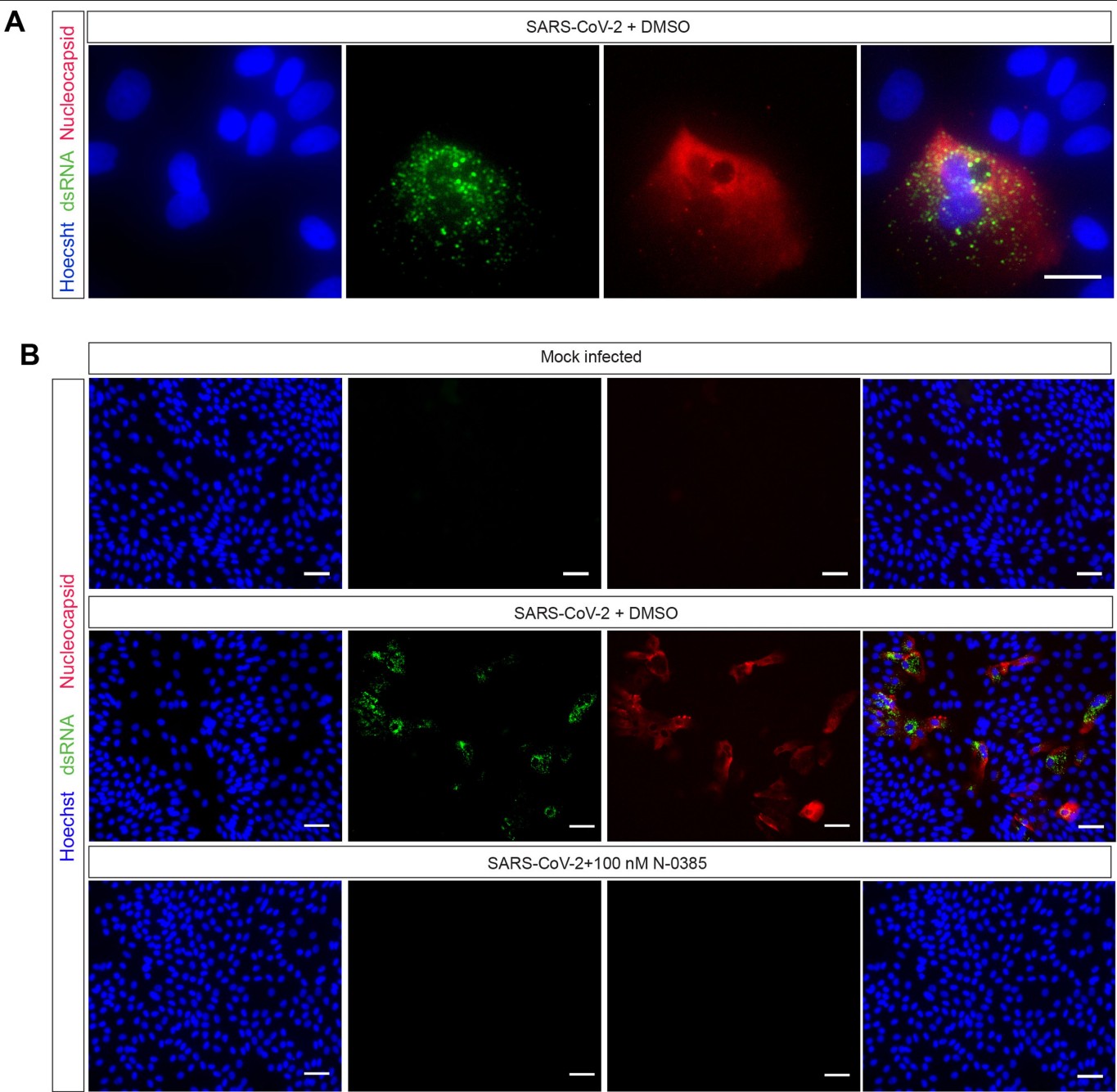

**Extended Data Fig. 4 | Representative fluorescent images of SARS-CoV-2-infected colonoids.** (**A**) Colonoids infected with SARS-CoV-2 (VIDO) + 0.1% DMSO are shown. Scale bar: 20 µm. (**B**) Mock, SARS-CoV-2 infected, and SARS-CoV-2 + 100 nM N-0385 treated colonoids are shown. Images in (**B**) represent Hoechst, dsRNA, Nucleocapsid and composite images presented in Fig. 2d. Scale bars are 50 µm. For (**A**) and (**B**) Hoechst is shown in blue, nucleocapsid in red and dsRNA in green. Images captured with EVOS M7000 Imaging System. The images are representative of n = 3 independent experiments.

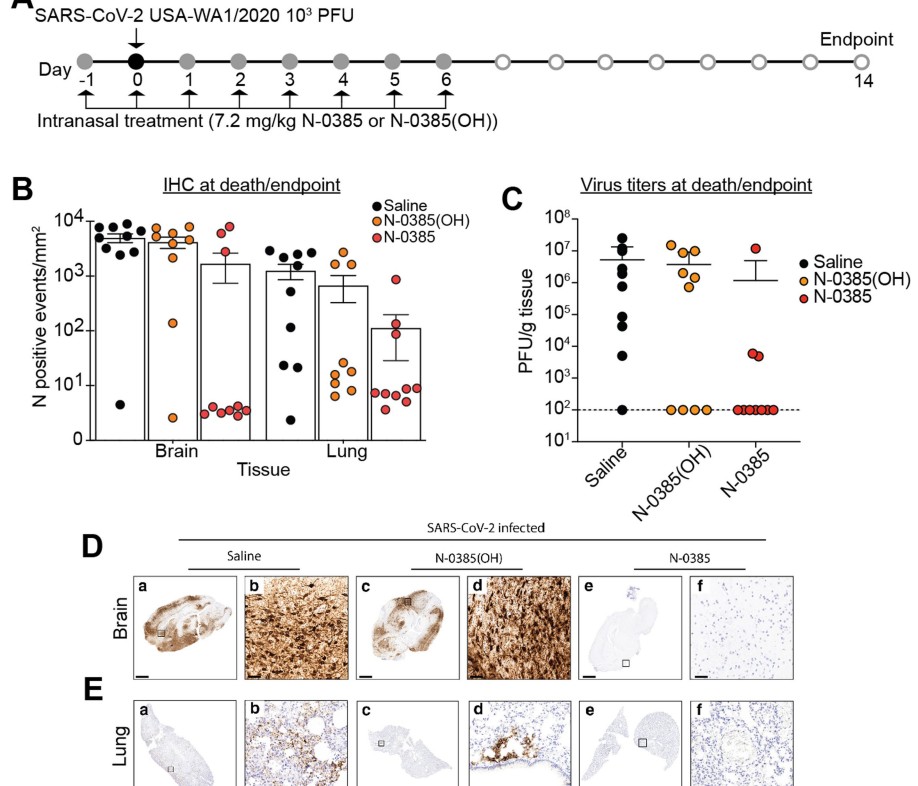

**Extended Data Fig. 5 | SARS-CoV-2 in the lungs of mice treated with N-0385 as demonstrated by IHC and plaque assay. (A)** Mice were treated daily on days −1 to +6 relative to challenge, with surviving mice terminated on Day 14 (same mice as Fig. 4). **(B)** Number of cells/mm² positive for SARS-CoV-2 nucleocapsid (N) at time of death or end-point by IHC staining. One whole lung slice evaluated per mouse. Data presented are mean ± SD **(C)** Virus titers (plaque-forming unit (PFU)/g of tissue) from the lungs of infected mice at time of death or end-point. Plaque assays were performed twice using a sample from each mouse and the average used to determine PFU/g. Data presented are mean ± SD **(D)** Representative sections of SARS-CoV-2 N staining in the brains of SARS-CoV-2 infected mice at time of death or end-point. Mice treated with saline (**a**, **b**: day 8) often had positive immunoreactivity in neurons throughout the brain. Immunoreactivity for SARS-CoV-2 was rare to absent in mice that

survived to the study end-point (**c**, **d**, **e**, **f**: day 14). **(E)** Representative sections of SARS-CoV-2 nucleocapsid in the lung of SARS-CoV-2-infected mice at time of death or end-point. Mice treated with saline (**a**, **b**: day 7) had immunoreactivity against SARS-CoV-2 throughout the lung. A similar pattern of patchy infection was present in mice treated with N-0385(OH) (**c**, **d**: day 6) but was not present in all mice. Immunoreactivity for SARS-CoV-2 was rare to absent in N-0385-treated mice that survived to the study end-point (**e**, **f**: day 14). Scale bar for **D**, **E**: **a**, **c**, **e** = 1 mm; **b**, **d**, **f** = 50 µm. For each experiment, 10 mice (5 males; 5 females) were analysed per treatment group. For histopathology and IHC analyses, representative images were selected based on the prevalent trend for a given treatment group, showing images representative of the mean pathological score.

**Extended Data Table 1 | Effective dose–response values ($IC_{50}$ and $EC_{50}$), cytotoxicity and selectivity index of lead peptidomimetic compounds against SARS-CoV-2 VIDO**

| Compound | Sequence | TMPRSS2 $IC_{50} \pm SD$ (nM) | SARS-CoV-2 VIDO Nucleocapsid $EC_{50} \pm SE$ (nM) | Calu-3 $CC_{50}$ (nM) | Selectivity Index |
|---|---|---|---|---|---|
| Camostat mesylate | $C_{21}H_{26}N_4O_8S$ | $17.5 \pm 18.8$ | $10.6 \pm 8.4$ | ND | ND |
| N-0130 | (H)RQFR-kbt | $3.1 \pm 1.5$ | $30.1 \pm 30.1$ | $2.7 \times 10^6$ | $8.97 \times 10^4$ |
| N-0438 | (H)REFR-kbt | $5.2 \pm 5.4$ | $35.7 \pm 24.5$ | $9.8 \times 10^5$ | $2.75 \times 10^4$ |
| N-0386 | Ac-QFR-kbt | $3.9 \pm 4.4$ | $2.3 \pm 1.7$ | $1.9 \times 10^6$ | $8.26 \times 10^5$ |
| N-0385 | Ms-QFR-kbt | $1.9 \pm 1.4$ | $2.8 \pm 1.4$ | $3.5 \times 10^6$ | $1.25 \times 10^6$ |
| N-0385(OH) | Ms-QFR-OH | $> 10000$ | $> 1000$ | ND | ND |

ND, not determined.

**Extended Data Table 2 | In vitro $K_i$ values calculated for the indicated proteases**

| Compound | $K_i$ (nM) | | | | | |
| --- | --- | --- | --- | --- | --- | --- |
| | Matriptase | Hepsin | DESC1 | Thrombin | Furin | Cathepsin L |
| N-0130 | 0.13 ± 0.03 | 0.38 ± 0.19 | 28 ± 10 | 8231 ± 1546 | > 10000 | > 10000 |
| N-0385 | 2.6 ± 0.4 | 0.57 ± 0.23 | 26.6 ± 8.8 | 9271 ± 647 | > 10000 | 6291 ± 3612 |
| N-0386 | 0.51 ± 0.09 | 0.60 ± 0.27 | 9.1 ± 1.9 | > 10000 | > 10000 | 7722 ± 142 |
| N-0438 | 1.11 ± 0.09 | 0.52 ± 0.07 | 31 ± 3 | > 10000 | > 10000 | > 10000 |
| Camostat mesylate | 5.6 ± 1.0 | 4.2 ± 1.6 | 5.8 ± 3.4 | 621 ± 46 | > 10000 | > 10000 |

Associated with Fig. 1d. Mean values ± s.d. ($n=3$, except cathepsin L versus N-0385 ($n=4$)).

## Extended Data Table 3 | Summary of histological findings

| Group | No. | Sex | DPI | Alveolar edema | Alveolar fibrin | Perivascular infiltrate in lung | Interstitial infiltrates in lung | Alveolar cells | Type II pneumocyte hyperplasia | Fibrosis | Lymphoid hyperplasia | Perivascular infiltrates in CNS | Necrotic cells in CNS | Gliosis in CNS | Meningeal infiltrate |
|---|---|---|---|---|---|---|---|---|---|---|---|---|---|---|---|
| Saline | 1 | F | 8 | 1 | 0 | 1 | 3 | 1 | 0 | 0 | 0 | 3 | 5 | + | + |
| | 2 | F | 6 | 1 | 1 | 3 | 1 | 1 | 0 | 0 | 0 | 3 | 3 | + | - |
| | 3 | F | 6 | 2 | 1 | 2 | 1 | 1 | 0 | 0 | 0 | 2 | 5 | + | - |
| | 4 | F | 6 | 0 | 1 | 2 | 1 | 1 | 1 | 0 | 0 | 2 | 0 | - | - |
| | 5 | F | 8 | 1 | 4 | 4 | 2 | 4 | 1 | 0 | 0 | 0 | 0 | - | - |
| | 6 | M | 7 | 3 | 2 | 3 | 1 | 2 | 1 | 0 | 0 | 1 | 1 | + | + |
| | 7 | M | 7 | 0 | 0 | 1 | 0 | 0 | 0 | 0 | 0 | 3 | 4 | + | + |
| | 8 | M | 7 | 1 | 1 | 2 | 0 | 0 | 0 | 0 | 0 | 1 | 2 | + | + |
| | 9 | M | 7 | 0 | 1 | 2 | 1 | 1 | 0 | 0 | 0 | 3 | 2 | + | + |
| | 10 | M | 6 | 1 | 1 | 4 | 2 | 3 | 1 | 0 | 0 | 1 | 0 | - | - |
| N-0385 (OH) | 11 | F | 6 | 0 | 0 | 1 | 1 | 0 | 0 | 0 | 0 | 0 | 0 | - | - |
| | 12 | F | 7 | 1 | 0 | 3 | 1 | 1 | 1 | 0 | 0 | 2 | 1 | - | - |
| | 13 | F | 7 | 0 | 0 | 1 | 0 | 0 | 0 | 0 | 0 | 3 | 3 | + | - |
| | 14 | F | 14 | 0 | 0 | 1 | 1 | 0 | 0 | 0 | 0 | 0 | 0 | - | - |
| | 15 | F | 14 | 0 | 0 | 1 | 0 | 0 | 0 | 0 | 0 | 1 | 0 | - | - |
| | 16 | M | 6 | 0 | 0 | 1 | 0 | 0 | 0 | 0 | 0 | 1 | 0 | - | - |
| | 17 | M | 6 | 1 | 1 | 3 | 1 | 1 | 1 | 0 | 0 | 0 | 0 | - | - |
| | 18 | M | 1 | 0 | 0 | 0 | 0 | 0 | 0 | 0 | 0 | 0 | 0 | - | - |
| | 19 | M | 9 | 2 | 2 | 3 | 3 | 2 | 3 | 0 | 0 | 3 | 2 | + | + |
| | 20 | M | 6 | 0 | 0 | 0 | 0 | 0 | 0 | 0 | 0 | 0 | 0 | - | - |
| N-0385 | 21 | F | 14 | 0 | 0 | 3 | 2 | 2 | 2 | 2 | 1 | 0 | 0 | - | - |
| | 22 | F | 14 | 0 | 0 | 3 | 1 | 1 | 1 | 1 | 1 | 0 | 0 | - | - |
| | 23 | F | 14 | 0 | 0 | 1 | 0 | 0 | 0 | 0 | 0 | 0 | 0 | - | - |
| | 24 | F | 14 | 0 | 0 | 0 | 0 | 0 | 0 | 0 | 0 | 0 | 0 | - | - |
| | 25 | F | 14 | 0 | 0 | 1 | 0 | 0 | 0 | 0 | 0 | 0 | 0 | - | - |
| | 26 | M | 14 | 0 | 0 | 3 | 1 | 1 | 3 | 1 | 0 | 0 | 0 | - | - |
| | 27 | M | 7 | 0 | 0 | 1 | 1 | 1 | 1 | 0 | 0 | 3 | 4 | + | + |
| | 28 | M | 8 | 0 | 0 | 0 | 0 | 0 | 0 | 0 | 0 | 1 | 2 | + | + |
| | 29 | M | 8 | 0 | 0 | 1 | 1 | 0 | 0 | 0 | 0 | 3 | 4 | + | + |
| | 30 | M | 14 | 0 | 0 | 1 | 1 | 0 | 1 | 0 | 0 | 0 | 0 | - | - |

Associated with Fig. 4. For lung, scores were applied based on the percentage of each tissue type (alveolus, vessels, etc.) affected using the following criteria: (0) normal, (1) <10% affected, (2) 10–25% affected, (3) 26–50% affected, and (4) > 50% affected. For brain, histological scoring was assessed for perivascular inflammation using the most severely affected vessel and the following criteria: (0) no perivascular inflammation, (1) incomplete cuff one cell layer thick, (2) complete cuff one cell layer thick, (3) complete cuff two to three cells thick, and (4) complete cuff four or more cells thick. Necrotic cells in the neuroparenchyma were assessed per $0.237\,mm^2$ field using the most severely affected area and the following criteria: (0) no necrotic cells (1) rare individual necrotic cells, (2) fewer than 10 necrotic cells, (3) 11 to 25 necrotic cells, (4) 26 to 50 necrotic cells, and (5) greater than 50 cells. DPI, days post-infection.

**Extended Data Table 4 | Summary of IHC detections using digital image analysis**

| | No. | DPI | Sex | Brain No. Detections | Brain No. Positive | Brain Positive % | Brain No. Positive per $mm^2$ | Brain Area $\mu m^2$ | Lung No. Detections | Lung No. Positive | Lung Positive % | Lung No. Positive per $mm^2$ | Lung Area $\mu m^2$ |
|---|---|---|---|---|---|---|---|---|---|---|---|---|---|
| Saline | 1 | 8 | F | 453970 | 223965 | 49.33 | 4974.6 | 45021429 | 301943 | 336 | 0.2314 | 23.29 | 30008571 |
| | 2 | 6 | F | 486359 | 374830 | 77.07 | 7773 | 48222090 | 400667 | 98447 | 24.57 | 2493.4 | 39483142 |
| | 3 | 6 | F | 465854 | 354560 | 76.11 | 7696.1 | 46069957 | 207789 | 62226 | 29.95 | 2896.7 | 21481719 |
| | 4 | 6 | F | 513887 | 453237 | 88.2 | 8910.8 | 50863741 | 255267 | 2896 | 1.134 | 115.17 | 25144846 |
| | 5 | 8 | F | 386549 | 171 | 0.0442 | 4.474 | 38217449 | 178717 | 27164 | 15.2 | 1545 | 17582272 |
| | 6 | 7 | M | 537250 | 128731 | 23.96 | 2410.9 | 53394910 | 390592 | 84093 | 21.53 | 2174.6 | 38670869 |
| | 7 | 7 | M | 408114 | 215853 | 52.89 | 5331.7 | 40484653 | 236916 | 55 | 0.0232 | 2.346 | 23445269 |
| | 8 | 7 | M | 451316 | 142848 | 31.65 | 3201.4 | 44620512 | 215343 | 456 | 0.2118 | 21.32 | 21389039 |
| | 9 | 7 | M | 429536 | 281501 | 65.54 | 6610.5 | 42583630 | 257917 | 13211 | 5.122 | 518.16 | 25495909 |
| | 10 | 6 | M | 355731 | 97118 | 27.3 | 2735.5 | 35503394 | 208496 | 54762 | 26.27 | 2657.2 | 20608928 |
| N-0385(OH) | 11 | 6 | F | 455410 | 205622 | 45.15 | 4555.9 | 45133288 | 147523 | 39230 | 26.59 | 2703.8 | 14509382 |
| | 12 | 7 | F | 392653 | 308647 | 78.61 | 7896.3 | 39087554 | 385579 | 61582 | 15.97 | 1622.2 | 37962415 |
| | 13 | 7 | F | 568631 | 423604 | 74.5 | 7494 | 56525916 | 220955 | 572 | 0.2589 | 26.11 | 21908738 |
| | 14 | 14 | F | 327808 | 84 | 0.0256 | 2.578 | 32589614 | 239373 | 191 | 0.0798 | 8.074 | 23655852 |
| | 15 | 14 | F | 370250 | 143563 | 38.77 | 3909.6 | 36720851 | 202231 | 315 | 0.1558 | 15.79 | 19949559 |
| | 16 | 6 | M | 399074 | 207074 | 51.89 | 5209.1 | 39752588 | 290098 | 315 | 0.1086 | 10.94 | 28784410 |
| | 17 | 6 | M | 456688 | 97743 | 21.4 | 2144.6 | 45575843 | 229152 | 37366 | 16.31 | 1638.9 | 22799350 |
| | 18 | 1 | M | 199686 | 210 | 0.1052 | 10.67 | 19687474 | 281521 | 98 | 0.0348 | 3.505 | 27961503 |
| | 19 | 9 | M | 297118 | 4043 | 1.361 | 137.97 | 29303917 | 215259 | 384 | 0.1784 | 17.97 | 21363623 |
| | 20 | 6 | M | 521415 | 310256 | 59.5 | 6010.9 | 51615283 | 282546 | 178 | 0.063 | 6.38 | 27900718 |
| N-0385 | 21 | 14 | F | 407628 | 113 | 0.0277 | 2.79 | 40507501 | 295210 | 192 | 0.065 | 6.545 | 29335514 |
| | 22 | 14 | F | 439501 | 131 | 0.0298 | 3.006 | 43573511 | 260794 | 131 | 0.0502 | 5.06 | 25889492 |
| | 23 | 14 | F | 585651 | 205 | 0.035 | 3.517 | 58293602 | 177531 | 152 | 0.0856 | 8.682 | 17507867 |
| | 24 | 14 | F | 611836 | 255 | 0.0417 | 4.198 | 60744423 | 186909 | 131 | 0.0701 | 7.096 | 18461399 |
| | 25 | 14 | F | 423553 | 133 | 0.0314 | 3.156 | 42136757 | 259815 | 95 | 0.0366 | 3.624 | 26215486 |
| | 26 | 14 | M | 497724 | 201 | 0.0404 | 4.077 | 49300400 | 370766 | 269 | 0.0726 | 7.345 | 36623344 |
| | 27 | 7 | M | 539950 | 429346 | 79.52 | 7967.3 | 53888613 | 231935 | 3067 | 1.322 | 133.68 | 22943342 |
| | 28 | 8 | M | 497665 | 137295 | 27.59 | 2771 | 49546591 | 193088 | 1648 | 0.8535 | 86.7 | 19007209 |
| | 29 | 8 | M | 459106 | 271623 | 59.16 | 5985.4 | 45380625 | 335357 | 28511 | 8.502 | 861.5 | 33094589 |
| | 30 | 14 | M | 360472 | 125 | 0.0347 | 3.495 | 35767364 | 308928 | 272 | 0.088 | 8.911 | 30522750 |

Associated with Extended Data Fig. 5d, e.

**Extended Data Table 5 | Summary of histological findings in the lung**

| Group | No. | Sex | DPI | Perivascular infiltrate in lung | Interstitial infiltrates in lung | Alveolar cells | Type II pneumocyte hyperplasia | Total Score |
|-------|-----|-----|-----|------|------|------|------|------|
| N-0385 | 1 | F | 3 | 1 | 1 | 0 | 0 | 2 |
| | 2 | F | 3 | 1 | 1 | 0 | 0 | 2 |
| | 3 | F | 3 | 1 | 1 | 0 | 1 | 3 |
| | 4 | F | 3 | 0 | 1 | 0 | 0 | 1 |
| | 5 | F | 3 | 0 | 1 | 0 | 0 | 1 |
| | 6 | M | 3 | 3 | 2 | 1 | 1 | 7 |
| | 7 | M | 3 | 0 | 1 | 0 | 0 | 1 |
| | 8 | M | 3 | 1 | 1 | 0 | 0 | 2 |
| | 9 | M | 3 | 1 | 1 | 0 | 0 | 2 |
| | 10 | M | 3 | 2 | 2 | 0 | 2 | 6 |
| Saline | 21 | F | 3 | 2 | 2 | 1 | 2 | 7 |
| | 22 | F | 3 | 1 | 1 | 0 | 1 | 3 |
| | 23 | F | 3 | 3 | 2 | 1 | 2 | 8 |
| | 24 | F | 3 | 1 | 0 | 0 | 0 | 1 |
| | 25 | F | 3 | 1 | 1 | 0 | 1 | 3 |
| | 26 | M | 3 | 2 | 2 | 1 | 2 | 7 |
| | 27 | M | 3 | 3 | 3 | 1 | 3 | 10 |
| | 28 | M | 3 | 1 | 1 | 1 | 1 | 4 |
| | 29 | M | 3 | 1 | 0 | 0 | 0 | 1 |
| | 30 | M | 3 | 2 | 2 | 2 | 2 | 8 |

Associated with Fig. 6g. Scores were applied based on the percentage of each tissue type (alveolus etc.) affected using the following criteria: (0) normal, (1) <10% affected, (2) 10–25% affected, (3) 26–50% affected, and (4) > 50% affected. Assessment was performed on tissue sections stained for IHC analysis.

# Reporting Summary

Nature Research wishes to improve the reproducibility of the work that we publish. This form provides structure for consistency and transparency in reporting. For further information on Nature Research policies, see our Editorial Policies and the Editorial Policy Checklist.

## Statistics

For all statistical analyses, confirm that the following items are present in the figure legend, table legend, main text, or Methods section.

| n/a | Confirmed | |
|---|---|---|
| ☐ | ☒ | The exact sample size (*n*) for each experimental group/condition, given as a discrete number and unit of measurement |
| ☐ | ☒ | A statement on whether measurements were taken from distinct samples or whether the same sample was measured repeatedly |
| ☐ | ☒ | The statistical test(s) used AND whether they are one- or two-sided *Only common tests should be described solely by name; describe more complex techniques in the Methods section.* |
| ☒ | ☐ | A description of all covariates tested |
| ☐ | ☒ | A description of any assumptions or corrections, such as tests of normality and adjustment for multiple comparisons |
| ☐ | ☒ | A full description of the statistical parameters including central tendency (e.g. means) or other basic estimates (e.g. regression coefficient) AND variation (e.g. standard deviation) or associated estimates of uncertainty (e.g. confidence intervals) |
| ☐ | ☒ | For null hypothesis testing, the test statistic (e.g. *F*, *t*, *r*) with confidence intervals, effect sizes, degrees of freedom and *P* value noted *Give P values as exact values whenever suitable.* |
| ☒ | ☐ | For Bayesian analysis, information on the choice of priors and Markov chain Monte Carlo settings |
| ☒ | ☐ | For hierarchical and complex designs, identification of the appropriate level for tests and full reporting of outcomes |
| ☒ | ☐ | Estimates of effect sizes (e.g. Cohen's *d*, Pearson's *r*), indicating how they were calculated |

*Our web collection on statistics for biologists contains articles on many of the points above.*

## Software and code

Policy information about availability of computer code

| Data collection | No software was used to collect data in this study |
|---|---|
| Data analysis | Molecular Operating Environment (MOE2019.01.02) from the Chemical Computing Group used to generate the TMPRSS2 model with N-0385; (HCS Studio Cell Analysis Software, version 4.0) was used to quantify viral markers of SARS-CoV-2 infection; The GraphPad Prism 9™ (GraphPad Software, Inc.) was used for statistical analysis; Digital image analysis for mouse/nucleocapsid staining was performed using QuPath software version 0.2.3; Leica Application Suite X (LAS X) software was used for acquiring confocal images. Merging of different channels from confocal imaging and the addition of the scale bar were performed using ImageJ/FIJI. |

For manuscripts utilizing custom algorithms or software that are central to the research but not yet described in published literature, software must be made available to editors and reviewers. We strongly encourage code deposition in a community repository (e.g. GitHub). See the Nature Research guidelines for submitting code & software for further information.

## Data

Policy information about availability of data

All manuscripts must include a data availability statement. This statement should provide the following information, where applicable:
- Accession codes, unique identifiers, or web links for publicly available datasets
- A list of figures that have associated raw data
- A description of any restrictions on data availability

All data generated or analyzed during this study are included in this published article (and its supplementary information files).

# Field-specific reporting

Please select the one below that is the best fit for your research. If you are not sure, read the appropriate sections before making your selection.

☒ Life sciences ☐ Behavioural & social sciences ☐ Ecological, evolutionary & environmental sciences

For a reference copy of the document with all sections, see nature.com/documents/nr-reporting-summary-flat.pdf

# Life sciences study design

All studies must disclose on these points even when the disclosure is negative.

| | |
|---|---|
| Sample size | Ten mice were used per group based on the following power analysis statistical power calculations: We sought a sample size that was large enough to detect significant differences in the EC50 averages (+/- standard deviations) between groups. We sought help from the Cornell Statistical Consulting Unit to perform our power analysis. For this, we used our pilot data and a 2-group, 2 sided analysis, setting our power (1-) to 0.9, to have 90% power to detect a statistical difference, and an error rate of 5%, to achieve a significance level of corresponding to a 5% error. These calculations resulted in an N number of animals per groups of seven. However, 10 animals were used to ensure statistical significance in the case of unexpected animal deaths.<br><br>No sample size calculations were performed for in vitro and cell culture experiments. Sample sizes were based on standards in the field, typically 3 independent biological experiments with each replicate assayed in technical triplicate or greater. Peptidomimetic screening (Fig 2a) against SARS-CoV-2 in Calu-3 was performed only twice (two independent experiments with 5 technical replicates each) to identify the lead compound for detailed downstream analysis. Real-time qPCR was only performed once to qualitatively verify the expression of ACE2 and TMPRSS2 in Calu-3 and colonoids. Due to limited sample availability, the control compound N-0385(OH) was only evaluated against SARS-CoV-2 twice in colonoids and the results were consistent in both experiments. Note that the active compounds N-0385 was tested independently three times in colonoids against SARS-CoV-2. |
| Data exclusions | 1 male animal was excluded from the N-0385(OH) control group that died of unknown causes (day -1-6 treatment), that did not affect statistical significance when excluded or included. GraphPad Prism was used to identify and eliminate outliers (Q = 1) for inhibitor screening, IC50 determination and protease selectivity (Ki). |
| Replication | Each experiment using 10 mice per treatment group (5 female and 5 male) was performed once. The number of animals used is sufficient to interpret the results with an error rate <5% using male and female animals precluding the necessity of conducting animal experiments in replicate. Also, each animal is considered an independent biological replicate. |
| Randomization | Mice were designated into groups randomly to reduce bias due to differences in weight, and animal studies were performed using age matched mice to compare across groups.<br><br>Randomization does not apply to cell-based and in vitro experiments, in which large numbers of cells or reagent aliquots from a given source are partitioned among experimental conditions. |
| Blinding | Investigators were not blinded to groups during weighing and scoring of animal health due to nature of the infectious agent but histopathology, IHC, and related data analyses were dealt with in a blinded fashion by certified pathologist and clinicians in the team.<br><br>Blinding was also not done for in vitro or cell-based experiments as no subjective rating of data was involved. Quantification was automated using plate readers, qPCR machines, high-content screening platforms. |

# Reporting for specific materials, systems and methods

We require information from authors about some types of materials, experimental systems and methods used in many studies. Here, indicate whether each material, system or method listed is relevant to your study. If you are not sure if a list item applies to your research, read the appropriate section before selecting a response.

## Materials & experimental systems

| n/a | Involved in the study |
|---|---|
| ☐ | ☒ Antibodies |
| ☐ | ☒ Eukaryotic cell lines |
| ☒ | ☐ Palaeontology and archaeology |
| ☐ | ☒ Animals and other organisms |
| ☒ | ☐ Human research participants |
| ☒ | ☐ Clinical data |
| ☐ | ☒ Dual use research of concern |

## Methods

| n/a | Involved in the study |
|---|---|
| ☒ | ☐ ChIP-seq |
| ☒ | ☐ Flow cytometry |
| ☒ | ☐ MRI-based neuroimaging |

# Antibodies

| | |
|---|---|
| Antibodies used | The SARS-CoV-2 nucleocapsid antibody [HL344] (GTX635679) was provided by Genetex; mouse anti-dsRNA antibody (J2-1904) was purchased from Scicons English and Scientific Consulting; Hoechst 33258 and secondary antibodies goat anti-mouse IgG Alexa Fluor 488 (A11001) and goat anti-rabbit IgG Alexa Fluor 555 (A21428) were obtained from Invitrogen |
| Validation | The nucleocapsid antibody was validated by the company (Genetex) using validation protocols in line with guidelines described by the International Working Group on Antibody Validation (IWGAV). Detailed validation and testing are available here: https://www.genetex.com/Product/Detail/SARS-CoV-2-COVID-19-nucleocapsid-antibody-HL344/GTX635679#datasheet. The manufacturer also states that "This antibody detects SARS-CoV-2 nucleocapsid protein, but does not cross-react with SARS-CoV or MERS-CoV nucleocapsid proteins based on our internal testing." <br> The dsRNA antibody was validated by SCICONS: https://www.jenabioscience.com/rna-technologies/rna-analysis-detection/dsrna-detection/rnt-sci-10010-anti-dsrna-monoclonal-antibody-j2. The manufacturer states: "Specificity: Anti-dsRNA monoclonal antibody J2 recognises double-stranded RNA (dsRNA) provided that the length of the helix is greater than or equal to 40 bp. dsRNA-recognition is independent of the sequence and nucleotide composition of the antigen. All naturally occurring dsRNAs investigated up to now (40-50 species) as well as poly(I)-poly(C) and poly(A)-poly(U) have been recognised by Anti-dsRNA monoclonal antibody J2 although in some assays its affinity to poly(I)-poly(C) is about 10 times lower than that to other dsRNA antigens." <br> Secondary antibodies and Hoecsht were validated by Invitrogen / Thermo Fisher <br> Both nucleocapsid and dsRNA antibodies were additionally verified for specificity using SARS-CoV-2 infected versus non-infected cells as shown in Extended Data Figure 2 and 4 (see mock vs. SARS-CoV-2 panels) where the antibody signal is not detected in non-infected / mock cells. |

# Eukaryotic cell lines

Policy information about cell lines

| | |
|---|---|
| Cell line source(s) | All cell lines (Calu-3 cells (ATCC® HTB-55™) and Vero E6 cells (ATCC® CRL-1586™)) were purchased from ATCC (American Type Culture Collection) |
| Authentication | None of the cell lines used were authenticated |
| Mycoplasma contamination | We confirm that all cell lines used tested negative for mycoplasma contamination |
| Commonly misidentified lines <br> (See ICLAC register) | No commonly misidentified cell lines were used in this study |

# Animals and other organisms

Policy information about studies involving animals; ARRIVE guidelines recommended for reporting animal research

| | |
|---|---|
| Laboratory animals | Eight-week-old heterozygous K18 hACE2 c57BL/6J male and female mice (strain: 2B6.Cg-Tg(K18-hACE2)2Prlmn/J) were obtained directly from or bred at Cornell using Jackson Laboratory obtained animals. The rodent housing setting includes a standard 12:12 dark/light cycle (for non-breeding rooms), with a temperature range of 68-72 F (20 to 22 °C), and a relative humidity in the range of 40-60%. |
| Wild animals | The study did not involve wild animals |
| Field-collected samples | No field collected samples were used in this study |
| Ethics oversight | Studies were approved and supervised following protocols set and reviewed under Institutional Animal Care and Use Committee at Cornell University (IACUC mouse protocol # 2017-0108 and BSL3 IBC # MUA-16371-1) |

Note that full information on the approval of the study protocol must also be provided in the manuscript.

# Dual use research of concern

Policy information about dual use research of concern

## Hazards

Could the accidental, deliberate or reckless misuse of agents or technologies generated in the work, or the application of information presented in the manuscript, pose a threat to:

| No | Yes | |
|---|---|---|
| ☐ | ☒ | Public health |
| ☐ | ☒ | National security |
| ☒ | ☐ | Crops and/or livestock |
| ☒ | ☐ | Ecosystems |
| ☒ | ☐ | Any other significant area |

| Hazards | Use of SARS-CoV-2 and SARS-CoV-2 variants of concern (risk group 3 agent) |
|---|---|

For examples of agents subject to oversight, see the United States Government Policy for Institutional Oversight of Life Sciences Dual Use Research of Concern.

## Experiments of concern

Does the work involve any of these experiments of concern:

| No | Yes | |
|---|---|---|
| ☒ | ☐ | Demonstrate how to render a vaccine ineffective |
| ☒ | ☐ | Confer resistance to therapeutically useful antibiotics or antiviral agents |
| ☒ | ☐ | Enhance the virulence of a pathogen or render a nonpathogen virulent |
| ☒ | ☐ | Increase transmissibility of a pathogen |
| ☒ | ☐ | Alter the host range of a pathogen |
| ☒ | ☐ | Enable evasion of diagnostic/detection modalities |
| ☒ | ☐ | Enable the weaponization of a biological agent or toxin |
| ☒ | ☐ | Any other potentially harmful combination of experiments and agents |

## Precautions and benefits

| Biosecurity precautions | All infections were carried out in a Biosafety Level 3 (BSL3) facility (UBC FINDER) in accordance with the Public Health Agency of Canada and UBC FINDER regulations (UBC BSL3 Permit # B20-0105 to FJ). Animal studies were performed under approved BSL-3 conditions and approved by the Institutional Animal Care and Use Committee at Cornell University (IACUC mouse protocol # 2017-0108 and BSL3 IBC # MUA-16371-1). All animal experiments were performed under USA CDC and USDA guidelines. |
|---|---|
| Biosecurity oversight | FINDER is UBC's shared platform for researchers working with Risk Group 3 pathogens. Bio-security oversight is provided by The FINDER Steering Committee members and the UBC Vice-President, Research and Innovation Group. All animal BSL-3 work at Cornell University is reviewed by both the IACUC and IBC committees. |
| Benefits | Benefit is that this work may mitigate risks to public health by providing new antiviral treatment options for SARS-CoV-2 associated disease (COVID-19) |
| Communication benefits | There are no risks to communicating this information |

