## [Peer Review File · Nature]

Manuscript Title: A TMPRSS2 inhibitor acts as a pan-SARS-CoV-2 prophylactic and therapeutic

Reviewer Comments & Author Rebuttals

Reviewer Reports on the Initial Version:

Referee #1 (Remarks to the Author):

Cell culture studies identified the cellular serine protease TMPRSS2 as an activator of the SARS-CoV-2 spike protein that is essential for efficient entry into human lung cells. Although findings supporting this scenario were made with cell lines and primary cells or lung tissue ex vivo, evidence for TMPRSS2 inhibitors being suitable for COVID-19 therapy is largely lacking. Here, Shapira and colleagues report the identification and characterization of a novel TMPRSS2 inhibitor, N-0385. In brief, they show that N-0385 inhibits TMPRSS2 with high potency (low nanomolar range) and efficiently blocks SARS-CoV-2 infection of Calu-3 lung cells and colonoids. Further, topical administration of N-0385 before, during and after infection resulted in a marked decrease of viral spread and weight loss of infected animals and a marked increase in survival. The study demonstrates that the generation of protease inhibitors active against TMPRSS2 is an attractive approach to COVID-19 prevention. These findings are novel and of significant interest to the field.

Major

It is important to demonstrate antiviral activity in primary human lung cells, using ALI cultures or related models.

Did the colonoids studied express TMPRSS2?

The formal demonstration that N-0385 inhibits murine TMPRSS2 seems to be lacking and should be provided.

Minor

The potential of N-0385 for human use should be discussed. Is the compound orally bioavailable, are safety issues expected upon IV application etc.?

„a 5-fold increase in fluorescent reporter activity“. This statement is unclear since TMPRSS2 WT has an estimated 95-fold higher activity as compared to TMPRSS2 with mutated active center, as judged from figure 1B.

Why did saline treatment induce additional histological changes?

Did the authors look for potential resistance development?

Legend to figure 5: "...mice from Figure 3F (a, b, g, h). There is no figure 3F.

Figure 4 F and G and figure 5 C and D: It should be indicated in the figure legend at what time after infection mice were sacrificed for analysis.

Analysis of viral genome copies or PFU in lung tissue would have strengthened the findings made with immunostaining/histology.

Referee #2 (Remarks to the Author):

In this manuscript, the authors screened and identified TMPRSS2 small molecule inhibitors for use as antiviral drugs against SARS-CoV-2. The inhibitors are tested in both Calu3 cells as well as colon organoid cell system. The inhibitor, N-0385, is also tested against B.1.1.7 and B.1.351 in Calu3 cells to determine if the compound inhibited the variants of concern similarly to SARS-CoV-2/WA1. The compounds were then tested in vivo using the K18/hACE2 mouse model and various dosing regimens to determine protection. The in vitro data demonstrates low nm IC50 and the mouse model demonstrates protection from disease. Questions concerning the mouse model, dosing and results remain.

1. Mice were dosed with 7.2mg/kg of N-0385 and N0385-OH. There doesn't seem to be any rationale for this dosing amount. Was PK performed and this dose identified as having high levels in the lungs of mice?

2. The dosing regimen in the mice needs some explanation. In Figure 4, mice were treated from day -1 to +6, and then the experiment was carried for 14 days total. There is no rationale given for the

timing of dosing. Please explain why dosing wasn't continued or these timepoints were taken. Is it toxicity of the compound past 6 days? For this experiment, virus titer in lung and brain is needed, as well as histological scoring of tissue sections in Figure 4.

3. In Figure 5, where IHC is performed, is this the same experiment as in Day 4 just analyzed for IHC or is this experiment repeated with another round of mice? It is unclear in the text. The IHC quantitation does not correlate with the IHC images. The images say they are representative, but the scoring for lungs shows low level positive staining while the images for saline and N0385-OH show very high IHC positive levels. Please explain or change the images to make them representative of the experiment. There are also black dots for sections scattered in the columns where there should only be brain from the N0385-OH treated mice and the lungs of the N0385 treated mice. Please correct the figure.

4. In Figure 6, a shorter timeframe of dosing is used with treatment from day -1 to +2. In this experiment the treated mice do better with a shorter treatment course than the Day -1 to +6 dosing in Figure 4, per weight loss and survival. There is no other metrics shown in this experiment and N0385-OH is left out of this experiment, so that no compound control is present. This experiment also does not have titer, pathology, IHC etc. It is unclear why this experiment is included and looks to not be a complete experiment. Please provide additional data to evaluate how a lower dose could protect mice.

5. For these experiments, a post infection treatment group is required to show usefulness as a therapeutic. Dosing at day +1 is needed.

6. To demonstrate the effects of the compound as a TMPRSS2 inhibitor, the authors should use a furin site mutant strain of SARS-CoV-2 and perform the in vitro inhibition experiments with it as compared to a wildtype strain to determine whether this inhibition is specific to cleavage of the furin cleavage site in Spike or other effects in the cell.

7. In the title of the paper, it refers to broadly protective in mice. I am assuming this is in relation to the VOC that are tested in vitro. The same VOC replicate well in the K18 mice and should be tested with this inhibitor in vivo.

Referee #3 (Remarks to the Author):

Shapira et al. describe a peptidic inhibitor, N-0385, of the type-II transmembrane serine protease, TMPRSS2, which is essential for cleavage and priming of the SARS-CoV-2 spike protein. N-0385 is mesylate-Gln-Phe-Arg-benzothiazole. The electron-withdrawing aromatic group primes the keto group of the C-terminal arginine of the compound for nucleophilic attack by the active-site serine residue of TMPRSS2. As a result, N-0385 is a covalently binding, reversible inhibitor of the protease, with an inhibition constant in the nanomolar range. ED50 values in virus-infected Vero E6 cells are

similar for the “wild-type” (presumably D614G) virus and for the alpha as well as beta variants of concern. There were no signs of toxicity; the selectivity index appears to be as high as 10(6).

The authors demonstrate that the more potent of the peptidic benzothiazoles inhibit SARS-CoV-2 entry in Calu3 cells and in donor-derived human colonoids. They further show that N-0385 is active in a COVID-19 animal model that employs hACE2 K18-mice (single daily intranasal administration). When applied early in infection (days -1 to 2), 100% of the mice survived. However, after an 8-day N-0385 treatment scheme ranging from day -1 to 7, only 70% of the mice were protected from SARS-CoV-2 mortality. I find this difficult to understand, because the compound shows no toxicity whatsoever. So elongated treatment should not do any harm, even though the mechanism of action of N-0385 is clearly connected to virus entry. Do the authors have an explanation for this?

Overall, the authors present novel data on a compound that seems to be superior to the known TMPRSS2 inhibitors camostat and nafamostat. Given the present emergence of SARS-CoV-2 variants of concern, the results are obviously of high significance. The data presented in this manuscript are of high quality, not the least because in all experiments, the authors used an inactive version of N-0385, which has its keto group reduced to hydroxyl, as a negative control. As far as I can see, appropriate use of statistics has been made. The conclusions made are well-justified on the basis of the experimental results. The presentation of the results is clear and the manuscript is well readable (a few language problems, mostly repetitions, are mentioned below).

Minor points:

In addition to my concern regarding the rate of protection against mortality, which I indicated above, I would like to mention a few minor points:

(1) line 33: The term “broad-spectrum coronavirus inhibitor” would normally indicate potency not only against SARS-like betacoronaviruses, but also alphacoronaviruses and other betacoronaviruses. I would not call activity against “wild-type” and two variants of the same virus “broad-spectrum”.

(2) line 63: coronavirus main protease. The abbreviation “Mpro” is introduced here, but throughout the rest of the manuscript, the alternative “3CLpro” is used...

(3) lines 71 – 73: This mouse model should be defined in a more elegant way (in terms of language).

(4) line 92: Here, the K18-hACE2 mouse model is introduced again (“a model of severe COVID19”), which is redundant after it has been described in lines 71 – 73.

(5) line 124: The amino-acid 3-letter symbol Cha should be explained at this point.

(6) lines 140 ff: The inactivity of N-0385 against thrombin is surprising. The non-peptidic TMPRSS2 inhibitor camostat mesylate is a pretty strong thrombin inhibitor. Do the authors have an explanation for the inactivity of their compound (perhaps by attempting to dock N-0385 into the thrombin structure)?

(7) line 151: A reference should be given for the crystal structure of matriptase (not just the PDB code).

(8) line 160: What is meant by the “glutamine ketone”? The oxygen of the side-chain amide, or the main-chain keto group?

(9) line 161: The hydrogen bond between the N-terminal mesylate and “the terminal amide of N-0385” is difficult to imagine, unless the authors mean the side-chain amide of the Gln residue... This should be clarified.

(10) line 168 and elsewhere in the manuscript: What are “small-molecule peptidomimetics”? Most peptidomimetics would be small, wouldn't they? I am not aware of “large-molecule peptidomimetics”.

(11) line 243: This sentence is repetitive.

(12) line 272: 20% survival

Referee #4 (Remarks to the Author):

- High level of need / timely study
- o So far we have no anti-viral drugs specifically targeted to COVID-19
- o Remdesivir has marginal efficacy in humans and requires IV access
- o Viral entry strategy has some efficacy for influenza -> reasonable approach
- o The work addresses a strong need (new therapeutics) and is novel (new target)
- Targeting a host protease required for infection has potential to evade resistance
- o TMPRSS2 is one such protease
- o TMPRSS2 required for second obligate cleavage of spike that is needed for fusion with cell membrane.
- o The work is well conceptualized for potential impact
- Building on prior work for influenza authors screen existing collection of TMPR inhibitors to find early lead.
- o Identify potent peptidomimetics (2-30 nM IC50); selective against small panel of related proteases
- o Compounds non-toxic to mammalian cells
- o Compounds block viral infection of Calu-3 cells (2-20 nM IC50)
- o Compounds block viral replication in Calu-3 cells (plaque assay) at 40 or 200 nM
- o Similar results found in primary colon cells that are normally infected
- o The compounds are good quality early leads with respect to potency, selectivity and early glimpse of toxicology
- o The compounds are uncharacterized with respect to pharmacokinetics and pharmaceutics – two critical aspects to demonstrate tractability for development.
- Focus on a single lead compound (N-0285)
- o Blocks variant forms of COVID-10 including Delta and Mu with similar potencies
- o Significantly reduces symptomology in humanized mouse model
- o The chosen lead shows efficacy in prophylactic models but no comment is made in treatment models. While clearly active against relevant viruses, it is hard to judge the applicability without showing an effective treatment of a challenge model.

Conclusions:

o Overall this is an interesting early phase study with reasonable characteristics for an early lead: good biochemical, cellular, in vivo activity with good correlations between the 3.

o The in vivo efficacy is shown in prophylactic mode studies and only in humanized mouse. The impact of the work would be much higher if efficacy were shown with treatment initiated at Day +1 or later. Additional impact would be obtained with a second animal model.

o The studies presented lack any pharmacokinetics. This makes it difficult to interpret the results in terms of exposure-efficacy relationships, although a total dose of 7.5 mg/kg is not unreasonable.

o The impact would be greatly enhanced if efficacy could be demonstrated from an oral route.

Author Rebuttals to Initial Comments:

Responses to reviewers' comments:

A. General comment received from Reviewer #1

This reviewer provided a very positive critique of our manuscript indicating that our study “demonstrates that the generation of protease inhibitors active against TMPRSS2 is an attractive approach to COVID-19 prevention” and that our findings are “novel and of significant interest to the field.”

Reviewer #1 (Remarks to the Author):

“Cell culture studies identified the cellular serine protease TMPRSS2 as an activator of the SARS-CoV-2 spike protein that is essential for efficient entry into human lung cells. Although findings supporting this scenario were made with cell lines and primary cells or lung tissue ex vivo, evidence for TMPRSS2 inhibitors being suitable for COVID-19 therapy is largely lacking. Here, Shapira and colleagues report the identification and characterization of a novel TMPRSS2 inhibitor, N-0385. In brief, they show that N-0385 inhibits TMPRSS2 with high potency (low nanomolar range) and efficiently blocks SARS-CoV-2 infection of Calu-3 lung cells and colonoids. Further, topical administration of N-0385 before, during and after infection resulted in a marked decrease of viral spread and weight loss of infected animals and a marked increase in survival. The study demonstrates that the generation of protease inhibitors active against TMPRSS2 is an attractive approach to COVID-19 prevention. These findings are novel and of significant interest to the field.”

A.1. Major points raised by Reviewer #1

(i) “It is important to demonstrate antiviral activity in primary human lung cells, using ALI cultures or related models”.

- *Response: In this revised manuscript, we have now demonstrated that N-0385 is a nanomolar antiviral against four SARS-CoV-2 VOCs (Alpha, Beta, Gamma, and Delta variants) in human lung cells (Calu-3 cells) (**new data**; Figure 3, panels A-D). Furthermore, we demonstrated the prophylactic and therapeutic efficacy of N-0385 against SARS-CoV-2 Delta variant in mice (**new data**; Figure 7, panels A-D)*

We believe the pan-SARS-CoV-2 antiviral activity of N-0385 with a selectivity index of $>1 \times 10^6$ in human Calu-3 cells with the demonstrated prophylactic and therapeutic efficacy against SARS-CoV-2 Delta variant in mice represent robust data both in cellulo and in vivo on the efficacy of N-0385 as a SARS-CoV-2 VOC antiviral.

Our findings are supported by a recent report by Mahoney et al. in PNAS that reports a similar peptidomimetic compound (MM3122) as a very potent inhibitor of recombinant TMPRSS2 activity that protects human epithelial lung cells from wild-type SARS-CoV-2 [Ref: Mahoney et al A novel class of TMPRSS2 inhibitors potently blocks SARS-CoV-2 and MERS-CoV viral entry and protects human epithelial lung cells, PNAS 2021].

(ii) “Did the colonoids studied express TMPRSS2?”

- *Response: We have performed QPCR in colonoids as well as Calu-3 cells to confirm the expression of TMPRSS2 and ACE2 and this has been added to the supplementary (**new data**; Figure S6).*

(iii) “The formal demonstration that N-0385 inhibits murine TMPRSS2 seems to be lacking and should be provided.”

- *Response: To address the reviewer's comment we have repeated the cellular TMPRSS2 reporter assay using murine TMPRSS2 and identified an IC_{50} of 12.3 ± 1.9 nM against the mouse enzyme (**new data**; Figure S3).*

A.2. Minor points raised by Reviewer #1

(i) “The potential of N-0385 for human use should be discussed. Is the compound orally bioavailable, are safety issues expected upon IV application etc.?”

- *Response: The objective for N-0385 use in humans is to maximize lung exposure to compounds while minimizing absorption and systemic exposure (although the consequences of systemic exposure need to be assessed). Therefore, airway administration was chosen for N-0385 and this class of compounds. Since TMPRSS2 is expressed in the human nose, olfactory epithelium, and olfactory bulb, and SARS-CoV-2 lodges in the nasal cavities and nasopharynx, even in asymptomatic or pre-symptomatic individuals, intranasal administration has several advantages for the prevention and treatment of SARS-CoV-2 and other viral diseases [Ref: Higgins TS et al. Intranasal Antiviral Drug Delivery and Coronavirus Disease 2019 (COVID-19): A State of the Art Review. Otolaryngology–Head and Neck Surgery. 2020;163(4):682-694].*

Several drugs are currently administered intranasally via a spray solution (allergic rhinitis, chronic rhinosinusitis, opioid overdose, and topical anesthesia/decongestion) or by nebulization (chronic rhinosinusitis and nasal polyposis) to treat a variety of long-term conditions and have been extensively reported in the literature. More recently in 2020, the FDA has approved Biohaven's intranasal vazegepant, a small-molecule, to enter Phase 2 clinical trials to blunt the severe inflammatory response in patients with COVID-19 on supplemental oxygen [Ref: <https://www.biohavenpharma.com/investors/news-events/press-releases/04-09-2020>]. Finally, preliminary results obtained by our group using nebulization showed that this class of compounds also reached the lungs in high quantities, between 6.01 and 12.9 µg/g in rats.

A similar peptidomimetic compound (MM3122) was recently reported to have “...excellent metabolic stability, safety, and pharmacokinetics in mice, with a half-life of 8.6 h in plasma and 7.5 h in lung tissue making it suitable for in vivo efficacy evaluation and a promising drug candidate for COVID-19 treatment; “MM3122 was administered daily to mice at three different single dose levels of 20, 50, and 100 mg/kg via IP injection over a period of 7 d. No adverse effects were observed in any of the treatment groups, and no weight loss or changes to harvested organs (liver, spleen, and kidneys) were noted compared to the control group (SI Appendix, Figure S5).” [Ref: Mahoney et al. A novel class of TMPRSS2 inhibitors potently block SARS-CoV-2 and MERS-CoV viral entry and protect human epithelial lung cells, PNAS 2021].

We have added this information to the discussion (Page 11, line 327 and Page 12, line 352-359)

(ii) “...a 5-fold increase in fluorescent reporter activity.” This statement is unclear since TMPRSS2 WT has an estimated 95-fold higher activity as compared to TMPRSS2 with mutated active center, as judged from figure 1B”

- *Response: Activity from mock-transfected cells was subtracted from all samples to eliminate background (which is approximately 20% of the signal in TMPRSS2-transfected cells) and then set to 100% in TMPRSS2-transfected cells (as described in figure legend). To avoid confusion, details were added in Materials and Methods and the sentence in the text was reworded: “Using this assay, we show that, as expected, the S441A substitution completely abrogated TMPRSS2 activity.” (page 5, line 113)*

(iii) “Why did saline treatment induce additional histological changes?”

- *Response: We stated in the results that “Saline-treated mice frequently had additional histological changes including alveolar edema, alveolar fibrin, and inflammatory cells within alveoli”. This statement refers to comparison with mice treated with N-0385; both infected with SARS-CoV-2. The additional histological changes are due to more severe SARS-CoV-2 infection and the lack of protection in untreated mice.
To clarify this point in the manuscript, the text has been reworded to: “Compared to N-0385-treated mice, control saline-treated mice frequently had additional histological changes [...]”
Page 9, line 259*

(iv) “Did the authors look for potential resistance development?”

- *Response: We did not look at resistance development due to safety concerns associated with potentially generating gain-of-function SARS-CoV-2 mutants with altered biological properties.*

(v) “Legend to figure 5: “...mice from Figure 3F (a, b, g, h). There is no figure 3F.”

- *Response: This has been corrected in the legend of Figure 5; the intention was to refer to Figure 4, for which the data in Figure 5 are matched. Page 40, line 979*

(vi) “Figure 4 F and G and figure 5 C and D: It should be indicated in the figure legend at what time after infection mice were sacrificed for analysis.”

- *Response: This information has been added to figure 4 and 5 legends. Page 39, lines 958 and 968; page 41, line 989-997*

(vii) “Analysis of viral genome copies or PFU in lung tissue would have strengthened the findings made with immunostaining/histology.”

- *Response: We have now performed a plaque assay from the lung tissue of mice treated with N-0385 and this has been added to Figure 5 and Figure 6, which excitingly shows a significant difference in the PFU levels in lung tissues +/- N-0385 treatment. (**New data**) Page 40 and 42.*

B. General comments received from Reviewer #2:

Reviewer #2 acknowledged that we have provided *in vitro* data for N-0385 that demonstrates low nm IC₅₀ and ED₅₀ values and that the mouse model demonstrates protection from disease. This reviewer has raised several points regarding the K18-hACE2 mouse model of severe human COVID-19 disease presented in our study; these are addressed in detail below.

Reviewer #2 (Remarks to the Author):

“In this manuscript, the authors screened and identified TMPRSS2 small molecule inhibitors for use as antiviral drugs against SARS-CoV-2. The inhibitors are tested in both Calu3 cells as well as colon organoid cell system. The inhibitor, N-0385, is also tested against B.1.1.7 and B.1.351 in Calu3 cells to determine if the compound inhibited the variants of concern similarly to SARS-CoV-2/WA1. The compounds were then tested *in vivo* using the K18/hACE2 mouse model and various dosing regimens to determine protection. The *in vitro* data demonstrates low nm IC₅₀ and the mouse model demonstrates protection from disease. Questions concerning the mouse model, dosing and results remain.”

B.1. Points raised by Reviewer #2

(i) “Mice were dosed with 7.2mg/kg of N-0385 and N0385-OH. There doesn't seem to be any rationale for this dosing amount. Was PK performed and this dose identified as having high levels in the lungs of mice? “

- *Response: The dosing of peptidomimetics was based on its therapeutic dose range as determined using preliminary studies with N-130 in mice infected with H1N1 and H3N2. The therapeutic dose range in these unpublished studies was determined to be 1-7.5 mg/kg based on morbidity, survival, and weight loss of mice. This dose was also chosen based on the solubility of the compounds in the volume of solution administered. To clarify this in our manuscript, dosing is now discussed in lines 245-248, page 9: “Dosing regimens and drug concentrations were chosen to maximize efficacy and were based on other studies in the K18-hACE2 mouse model and on the solubility of N-0385. The duration was chosen based on knowledge that untreated infected K18-hACE2 mice typically survive 6 to 8 days post-infection with SARS-CoV-2”*

(ii) “The dosing regimen in the mice needs some explanation. In Figure 4, mice were treated from day -1 to +6, and then the experiment was carried for 14 days total. There is no rationale given for the timing of dosing. Please explain why dosing wasn't continued or these timepoints were taken. Is it toxicity of the compound past 6 days? “

- *Response: The dosing regimen was initially chosen to maximize the therapeutic effect of the compound and was also influenced based on the known course of COVID-19 in the K18-mouse model, which typically survive 6-8 days post-infection when infected with 10³ PFU. We chose once/daily dosing to minimize the handling and isoflurane treatment (anesthesia) of infected mice. This is now added to the discussion in line 303 (Refs: Winkler, E. S. et al. SARS-CoV-2 infection of human ACE2-transgenic mice causes severe lung inflammation and impaired function. Nat. Immunol. 21, 1327–1335 (2020) and Yinda, C. K. et al. K18-hACE2 mice develop respiratory disease resembling severe COVID-19. PLOS Pathogens. 17, e1009195 (2021)].*

(iii) “sections in Figure 4”

Response: The histological scoring of tissue sections is available as a table in supplement (Table S3) and we have now added the virus titer from lung to Figure 4 and Figure 5. (New data) Page 40 and 42.

- Responses to reviewers' comments -

(iv) “In Figure 5, where IHC is performed, is this the same experiment as in Day 4 just analyzed for IHC or is this experiment repeated with another round of mice? It is unclear in the text.”

- *Response: Figure 5 is paired with Figure 4, so that the same animals were analyzed for both figures. This has now been clarified in the legend of Figure 5. Page 40, line 979*

(v) “The IHC quantitation does not correlate with the IHC images. The images say they are representative, but the scoring for lungs shows low level positive staining while the images for saline and N0385-OH show very high IHC positive levels. Please explain or change the images to make them representative of the experiment.”

- *Response: This is an excellent point and we have now edited the figure to choose one image that is the most representative of the average IHC levels for each group of 10 mice. Figure 5, page 40*

(vi) “There are also black dots for sections scattered in the columns where there should only be brain from the N0385-OH treated mice and the lungs of the N0385 treated mice. Please correct the figure.”

- *Response: This was an error incorporated during figure editing and has now been corrected. Figure 5, page 40*

(vii) “In Figure 6, a shorter timeframe of dosing is used with treatment from day -1 to +2. In this experiment the treated mice do better with a shorter treatment course than the Day -1 to +6 dosing in Figure 4, per weight loss and survival. There is no other metrics shown in this experiment and N0385-OH is left out of this experiment, so that no compound control is present. This experiment also does not have titer, pathology, IHC etc. It is unclear why this experiment is included and looks to not be a complete experiment. Please provide additional data to evaluate how a lower dose could protect mice.”

- *Response: Although the N-0385-treated mice appear to perform better with the shorter duration, please note that the saline mice also do better in this experiment; thus, the relative protection afforded by N-0385 compared to saline-treated mice is similar between both dosing regimens. The point of the shorter dose regimen duration experiment in Figure 5, as well as the even shorter new dose regimen duration experiments now in Figures 6 and 7 are to investigate how short of a dose regimen is sufficient to provide a prophylactic or therapeutic benefit to the animal upon SARS-CoV-2 infection. Our understanding is that the differences are due to experimental variation / heterogeneity in mouse survival outcomes across experiments, but the relative benefits between experiments in Figures 4/5 and 6 are consistent.*

(viii) “For these experiments, a post infection treatment group is required to show usefulness as a therapeutic. Dosing at day +1 is needed.”

- *Response: Performing day +1 experiments in this mouse model is challenging due to rapid spread of SARS-CoV-2 infection to the brain. However, to provide evidence of the therapeutic potential of this compound, we tested treatment at 0 hpi or 12 hpi, using the SARS-CoV-2 Delta VOC, and observed protection against weight loss at 6 dpi in both cases. Future studies will examine the therapeutic potential further in other animal models and with other doses regimens of compound. (New data) Page 43, Figure 7 and Page 10, line 284-296*

(ix) “To demonstrate the effects of the compound as a TMPRSS2 inhibitor, the authors should use a furin site mutant strain of SARS-CoV-2 and perform the in vitro inhibition experiments with it as compared to a wildtype strain to determine whether this inhibition is specific to cleavage of the furin cleavage site in Spike or other effects in the cell.”

- *Response: We have demonstrated that N-0385 does not inhibit furin enzymatic activity as shown in Figure 1D and Table S2. The compound is unlikely to impact furin activity as it has a different substrate recognition profile than that which is present in N-0385. Also, see response to Reviewer 3 (vii) (page 8) regarding modelling of the interaction between N-0385 and furin.*

- Responses to reviewers' comments -

(x). “In the title of the paper, it refers to broadly protective in mice. I am assuming this is in relation to the VOC that are tested in vitro. The same VOC replicate well in the K18 mice and should be tested with this inhibitor in vivo”.

- *Response: We have now performed two in vivo studies looking at the therapeutic efficacy of N-0385 against SARS-CoV-2 Delta variant in the K18-hACE2 mouse model of severe human COVID-19 disease (presented in Figure 7, page 43).*

C. General comments received from Reviewer #3

This reviewer provided a very detailed and positive critique of our manuscript indicating that “the results are high significance”, the data presented in our manuscript “are of high quality”, and the conclusions made are “well-justified on the basis of the experimental results”. This reviewer underlined the importance of our inactive version of N-0385, as a negative control in our study for the robust interpretations of the results of our studies both *in cellulo* and *in vivo*. This reviewer has raised several minor points that are addressed in detail below.

Reviewer #3 (Remarks to the Author):

“Overall, the authors present novel data on a compound that seems to be superior to the known TMPRSS2 inhibitors camostat and nafamostat. Given the present emergence of SARS-CoV-2 variants of concern, the results are obviously of high significance. The data presented in this manuscript are of high quality, not the least because in all experiments, the authors used an inactive version of N-0385, which has its keto group reduced to hydroxyl, as a negative control. As far as I can see, appropriate use of statistics has been made. The conclusions made are well-justified on the basis of the experimental results. The presentation of the results is clear and the manuscript is well readable (a few language problems, mostly repetitions, are mentioned below). Shapira et al. describe a peptidic inhibitor, N-0385, of the type-II transmembrane serine protease, TMPRSS2, which is essential for cleavage and priming of the SARS-CoV-2 spike protein. N-0385 is mesylate-Gln-Phe-Arg-benzothiazole. The electron-withdrawing aromatic group primes the keto group of the C-terminal arginine of the compound for nucleophilic attack by the active-site serine residue of TMPRSS2. As a result, N-0385 is a covalently binding, reversible inhibitor of the protease, with an inhibition constant in the nanomolar range. ED50 values in virus-infected Vero E6 cells are similar for the “wild-type” (presumably D614G) virus and for the alpha as well as beta variants of concern. There were no signs of toxicity; the selectivity index appears to be as high as 10(6).”

C.1. Points raised by Reviewer #3

(i) “The authors demonstrate that the more potent of the peptidic benzothiazoles inhibit SARS-CoV-2 entry in Calu3 cells and in donor-derived human colonoids. They further show that N-0385 is active in a COVID-19 animal model that employs hACE2 K18-mice (single daily intranasal administration). When applied early in infection (days -1 to 2), 100% of the mice survived. However, after an 8-day N-0385 treatment scheme ranging from day -1 to 7, only 70% of the mice were protected from SARS-CoV-2 mortality. I find this difficult to understand, because the compound shows no toxicity whatsoever. So elongated treatment should not do any harm, even though the mechanism of action of N-0385 is clearly connected to virus entry. **Do the authors have an explanation for this?**”

- *Response: Although the N-0385-treated mice appear to perform better with the shorter duration, please note that the saline mice also do better in this second experiment, thus the relative protection afforded by N-0385 compared to saline-treated mice is similar between both dosing regimens. Our understanding is that the difference is due to experimental variation / heterogeneity in mouse survival outcomes across the two experiments, but the relative benefits in both experiments are consistent.*

(ii) “line 33: The term “broad-spectrum coronavirus inhibitor” would normally indicate potency not only against SARS-like betacoronaviruses, but also alphacoronaviruses and other betacoronaviruses. I would not call activity against “wild-type” and two variants of the same virus “broad-spectrum”. ”

- *Response: We have removed the terminology of broad-spectrum inhibitor throughout the manuscript.*

- Responses to reviewers' comments -

(iii) “line 63: coronavirus main protease. The abbreviation “Mpro” is introduced here, but throughout the rest of the manuscript, the alternative “3CLpro” is used...”

- *Response: The wording has been changed to introduce the enzyme as 3CLpro. Page 3, line 58*

(iv) “lines 71 – 73: This mouse model should be defined in a more elegant way (in terms of language)”

- *Response: We have edited the wording of this sentence. Page 4, line 77*

(v) “line 92: Here, the K18-hACE2 mouse model is introduced again (“a model of severe COVID19”), which is redundant after it has been described in lines 71 – 73”.

- *Response: We have deleted the redundant part of the sentence. Page 4, line 86*

(vi) “line 124: The amino-acid 3-letter symbol Cha should be explained at this point.”

- *Response: edited Page 5, line 121*

(vii) “lines 140: The inactivity of N-0385 against thrombin is surprising. The non-peptidic TMPRSS2 inhibitor camostat mesylate is a pretty strong thrombin inhibitor. Do the authors have an explanation for the inactivity of their compound (perhaps by attempting to dock N-0385 into the thrombin structure)?”

- *Response: Our findings are consistent with a recent report by Mahoney et al. that did not detect inhibitory activity against thrombin for camostat or peptidomimetic inhibitors targeted at TTSPs. [Ref: Mahoney et al. A novel class of TMPRSS2 inhibitors potently block SARS-CoV-2 and MERS-CoV viral entry and protect human epithelial lung cells, PNAS 2021]*

In addition, as shown in the figure below, the warhead (benzothiazole) binding site is narrower for thrombin and furin than for TMPRSS2. Moreover, another major difference comes from the fact that this pocket is much more hydrophobic for TMPRSS2 (yellow circle). Indeed, the upper section being very polar for thrombin and furin, it cannot correctly accommodate the aromatic ring of benzothiazole, which would imply a high energetic cost of desolvation and unfavourable interactions, which is not thermodynamically favourable. Moreover, for thrombin, in order to adequately accommodate the binding site, the angle between the warhead and the arginine in P1 must be lower, which is impossible given the rigidity of the system. Also, an important steric clash with the phenylalanine side chain is shown in Figures 1 c) and d) in the thrombin binding site.

Figure 1: Docking of N-0385 in a) TMPRSS2 (van der Waals (VDW) interaction) b) TMPRSS2 (molecular surface), c) Thrombin (VDW interaction) d) Thrombin (molecular surface), e) Furin (VDW interaction) f) Furin (molecular surface).

(viii) “line 151: A reference should be given for the crystal structure of matriptase (not just the PDB code)”

- *Response: A reference has been added. Page 6, line 149*

(ix) “line 160: What is meant by the “glutamine ketone”? The oxygen of the side-chain amide, or the main-chain keto group? “

- *Response: The text has been changed to “molecule to the oxygen of the main-chain ketone group.” Page 6, line 157*

(x) “line 161: The hydrogen bond between the N-terminal mesylate and “the terminal amide of N-0385” is difficult to imagine, unless the authors mean the side-chain amide of the Gln residue... This should be clarified”

- *Response: The text has been changed to “with the side-chain amide of the Gln residue of N-0385.” Page 6, line 159*

(xi) “line 168 and elsewhere in the manuscript: What are “small-molecule peptidomimetics”? Most peptidomimetics would be small, wouldn’t they? I am not aware of “large-molecule peptidomimetics”.”

- *Response: edited throughout the text*

(xii) “line 243: This sentence is repetitive.”

- *Response: Repeated statement has been deleted.*

(xiii) “line 272: 20% survival”

- *Response: The spelling has been corrected Page 10, line 278*

D. General comment received from Reviewer #4

This reviewer provided a very positive critique of our manuscript indicating that our study is “timely” and respond to “High level of need” (“new therapeutics & “novel (new target)”. This reviewer also acknowledged that our work is “well conceptualized for potential impact”.

Reviewer #4 (Remarks to the Author):

“Overall this is an interesting early phase study with reasonable characteristics for an early lead: good biochemical, cellular, in vivo activity with good correlations between the 3. The in vivo efficacy is shown in prophylactic mode studies and only in humanized mouse. The impact of the work would be much higher if efficacy were shown with treatment initiated at Day +1 or later”.

“High level of need / timely study; The work addresses a strong need (new therapeutics) and is novel (new target; Targeting a host protease required for infection has potential to evade resistance; TMPRSS2 is one such protease; The work is well conceptualized for potential impact; Identify potent peptidomimetics (2-30 nM IC50); selective against small panel of related proteases; Compounds non-toxic to mammalian cells; Compounds block viral infection of Calu-3 cells (2-20 nM IC50); Compounds block viral replication in Calu-3 cells (plaque assay) at 40 or 200 nM; Similar results found in primary colon cells that are normally infected; The compounds are good quality early leads with respect to potency, selectivity and early glimpse of toxicology”.

D.1. Points raised by Reviewer #4

(i) “The compounds are uncharacterized with respect to pharmacokinetics and pharmaceutics – two critical aspects to demonstrate tractability for development.”

- *Response: A similar peptidomimetic compound (MM3122) was recently reported to have “...excellent metabolic stability, safety, and pharmacokinetics in mice, with a half-life of 8.6 h in plasma and 7.5 h in lung tissue making it suitable for in vivo efficacy evaluation and a promising drug candidate for COVID-19 treatment.”*
“MM3122 was administered daily to mice at three different single dose levels of 20, 50, and 100 mg/kg via IP injection over a period of 7 d. No adverse effects were observed in any of the treatment groups, and no weight loss or changes to harvested organs (liver, spleen, and kidneys) were noted compared to the control group (SI Appendix, Fig. S5).”
[Ref: Mahoney et al A novel class of TMPRSS2 inhibitors potently blocks SARS-CoV-2 and MERS-CoV viral entry and protects human epithelial lung cells, PNAS 2021].

We have added this information to the discussion. Page 11, line 327

(ii) “The in vivo efficacy is shown in prophylactic mode studies and only in humanized mouse. The impact of the work would be much higher if efficacy were shown with treatment initiated at Day +1 or later.”

- *Response: Performing day +1 experiments in this mouse model is challenging due to rapid spread of SARS-CoV-2 infection to the brain. However, to provide evidence of the therapeutic potential of this compound, we tested treatment at 0 hpi and 12 hpi, using infection with SARS-CoV-2 Delta VOC, and observed protection against weight loss at 6 dpi in both cases. Future studies will examine the therapeutic potential further in other animal models. (New data) Page 43, Figure 7 and Page 10, line 284-296*

- Responses to reviewers' comments -

(iii) “Additional impact would be obtained with a second animal model.”

- *Response: We agree that future studies should explore the efficacy of N-0385 in other animal models. Due to the need for timely publication and dissemination of our important discovery for developing novel COVID-19 therapeutics, we opted to complete studies in this established model of severe human COVID-19 disease, and future work will explore N-0385 treatment in other animal models such as the hamster model infected with SARS-CoV2 VOCs.*

(iv) “The studies presented lack any pharmacokinetics. This makes it difficult to interpret the results in terms of exposure-efficacy relationships, although a total dose of 7.5 mg/kg is not unreasonable.”

- *Response: See response to (i) above.*

(v) “The impact would be greatly enhanced if efficacy could be demonstrated from an oral route.”

- *Response: The objective for N-0385 use in humans is to maximize lung exposure to compounds while minimizing absorption and systemic exposure (although the consequences of systemic exposure need to be assessed). Therefore, airway administration was chosen for N-0385 and this class of compounds. Since TMPRSS2 is expressed in human nose, olfactory epithelium, and olfactory bulb and SARS-CoV-2 lodges in the nasal cavities and nasopharynx, even in asymptomatic or pre-symptomatic individuals, intranasal administration has several advantages for the prevention and treatment of SARS-CoV-2 and other viral diseases. [Higgins TS, Wu AW, Illing EA, et al. Intranasal Antiviral Drug Delivery and Coronavirus Disease 2019 (COVID-19): A State of the Art Review. Otolaryngology–Head and Neck Surgery. 2020;163(4):682-694].*

Discussed on line 352-359, page 12: “Intranasal administration has several advantages for the prevention and treatment of SARS-CoV-2 and other viral diseases, including ease of self-administration. SARS-CoV-2 mainly enters the human body through ACE2 and TMPRSS2 positive nasal epithelial cells. Intranasal drug delivery maximizes airway and lung exposure while limiting systemic exposure. For example, intranasal administration of a membrane fusion inhibitory lipopeptide prevented transmission of SARS-CoV-2 in ferrets; however, efficacy of intranasal delivery of a small molecule inhibitor has not been shown. Under our conditions, intranasal administration”

Reviewer Reports on the First Revision:

Referee #1 (Remarks to the Author):

The authors have adequately addressed all points raised by this reviewer. The inclusion of VOC data obtained in vitro and B.1.617.2 data obtained in the mouse model further strengthened the manuscript.

Referee #2 (Remarks to the Author):

The authors have significantly improved the manuscript from the initial submission adding more data and discussion. There still remains several questions concerning initial comments.

1. In vivo challenge experiment and dosing: In the challenge experiments in Figure 4, 5 and 6, it is unclear what timepoint the titers and IHC are analyzed. The text says at experiment endpoint but the treated mice live for 14 days while the untreated mice die earlier. Are the lungs of these mice analyzed at different days for each group? It would seem that the data is unmatched timepoints, although this is unclear in the text. In Figure 6 there are 2 mice that survive the full 14 days in the Saline group and 2 mice that have titer at the limit of detection in 6E. Are these the 2 mice that survive the whole saline only treatment? More clarity needs to be stated in the text for timing of these experiments for both lung titer and IHC

If different timepoints for the end of the experiment were used, then they cannot be compared to each other. Lung titers and IHC need to be analyzed at the same timepoint in the experiment.

2. In the new Figure 7 where 12 hr post exposure was tested with a single dose. The experimental result requires titer and lung pathology analysis, as performed in the other experiments.

3. My previous comments on PK and dosing regimen were not answered. It is unclear what “chosen to maximize efficacy” means. If maximizing efficacy was the goal then treatment each day of the experiment should have been conducted. The citation of other peptidomimetics on lines 327-331 does not mean that this compound will act the same.

Referee #3 (Remarks to the Author):

It seems the authors have dealt with the concerns I had raised.

Referee #4 (Remarks to the Author):

Overall the authors have done a reasonable job of addressing my earlier critiques, questions, and suggestions. In particular addition of two additional dose initiation points that are post infection alleviates some concerns about the approach. Overall the manuscript is acceptable in its current form.

Author Rebuttals to First Revision:

Reviewer #2 (Remarks to the Author):

Comment 1A:

“In vivo challenge experiment and dosing: In the challenge experiments in Figure 4, 5 and 6, it is unclear what timepoint the titers and IHC are analyzed. The text says at experiment endpoint but the treated mice live for 14 days while the untreated mice die earlier. Are the lungs of these mice analyzed at different days for each group? It would seem that the data is unmatched timepoints, although this is unclear in the text. More clarity needs to be stated in the text for timing of these experiments for both lung titer and IHC”

***Response:** We thank the reviewer for these excellent questions and suggestions to improve our manuscript. In the initial in vivo experiment presented in Figure 4, the primary goal was to test the survival benefit of N-0385 against COVID-19. Using the tissue samples from this survival experiment, which were obtained at the time of death (day 6-9) or at the study endpoint (day 14), we observed lower average viral titers and antigen staining in N-038 -treated mice compared to controls.*

As requested by the reviewer, we have added additional clarity to the text, figure legends, and panels to indicate when titers and IHC were analyzed:

- 1. Example (line 260-261): “In this first in vivo experiment, histological examination of lung tissue obtained either at time of death (6-9 dpi) or study endpoint (14 dpi) ...”*
- 2. Example (line 272-273): “Immunohistochemistry (IHC) of the SARS-CoV-2 nucleocapsid protein and plaque assay from the tissues harvested at time of death (6-9 dpi) or endpoint (14 dpi) revealed”*
- 3. Example new Figure 5H legend (line 1010): “Virus titers (PFU/g of tissue) from the lungs of infected mice at the time of death or study endpoint.”*
- 4. Example new Figure 5H heading: “Virus titers at death/endpoint”*

Comment 1B:

“In Figure 6 there are 2 mice that survive the full 14 days in the Saline group and 2 mice that have titer at the limit of detection in 6E. Are these the 2 mice that survive the whole saline only treatment? “

***Response:** Yes, these mice that survive are the ones that also clear infection. A clarifying sentence has been added to the manuscript line 290: “...the two saline treated mice that survived to the endpoint”*

Comment 1C: “If different timepoints for the end of the experiment were used, then they cannot be compared to each other. Lung titers and IHC need to be analyzed at the same timepoint in the experiment.”

Response: *We acknowledge the reviewer’s concern about comparing samples taken at different time points and thus have removed the statistical analysis comparing these samples and moved Figure 5 to the supplementary section as Figure S8. The data may be of interest to the readers, as they demonstrate that mice that survive to the study endpoint clear detectable infection, while those that die early on have not cleared infection. More importantly, to further address the reviewer’s concern and to fairly compare viral loads at the same time point, we expanded the data available for the experiment presented in **new Figure 5 (formerly Figure 6)**. The new data compare viral titers and IHC staining from lung tissue collected at the same time point: 3 days post-infection (dpi). This is provided in addition to viral titers analyzed at the study endpoint. The new data are added to Figure 5 F-G and text line 288-289: “Our analysis of viral loads via plaque assays and IHC at 3 dpi demonstrated that N-0385 treated mice had significantly reduced viral titers (97%) and IHC staining (98%), compared to control saline-treated mice”.*

Further, we added a direct comparison of weight loss experienced on 6 dpi by all groups (time point chosen because all mice in our studies survived to 6 dpi), presented in Figures 4E and 5E. We added to line 254-255: “At 6 dpi (before any mice had died) the saline control and N-0385(OH) treated mice lost on average 14% and 12% of their weight, respectively, while N-0385 treated mice lost on average only 3% weight” and to line 284-285: “At 6 dpi (before any mice had died) the saline control mice lost on average 10% of their weight, while N-0385 treated mice gained 1% weight.”.

Comment 2:

“In the new Figure 7 where 12 hr post exposure was tested with a single dose. The experimental result requires titer and lung pathology analysis, as performed in the other experiments.”

Response: *We have repeated the 12 hour-post infection (dpi) experiment to add the data requested by the reviewer. The mice lung tissues were harvested 3 dpi with SARS-CoV-2 Delta; titers, immunohistochemistry (IHC) staining, and pathology scores were compared. The data are presented in **new Figure 6 (previously Figure 7)**, panels E, F, and G, and show a significant reduction in viral titer and IHC staining (>50%) at this time point in N-0385-treated mice compared to controls infected with SARS-CoV-2 Delta. There was a 1.9-fold (46%) improvement in pathology scores although this did not reach significance (p-value 0.056). Line 305-309: “mice were treated with N-0385 or saline control at 12 hpi and lung tissue was harvested 3 dpi for plaque assays and IHC to measure viral titers and nucleocapsid staining (Figure 6E-F). This demonstrated >50% reduction in viral titers and IHC viral staining in N-0385-treated mice compared to the control saline group. Similarly, total pathology scores of lung tissue assessed using IHC sections were improved by approximately 1.9-fold (or 46%).”*

Comment 3:

“My previous comments on PK and dosing regimen were not answered. It is unclear what “chosen to maximize efficacy” means. If maximizing efficacy was the goal, then treatment each day of the experiment should have been conducted. The citation of other peptidomimetics on lines 327-331 does not mean that this compound will act the same.”

Response: *We agree with the reviewer’s concern and provide further answers below.*

The preliminary dosing regimens were originally chosen based on information collected during previous unpublished in vivo mouse studies performed with influenza A virus; these studies will be published in a separate manuscript, but we have provided the information below (Appendix A) for the reviewer.

As stated in the text (line 328), influenza A virus also requires TTSP cleavage of the viral glycoprotein for entry and we have previously published the anti-influenza activity of peptidomimetics targeting TTSPs (Beaulieu et al. 2013)

[REDACTED]

-
-

The statement in the text has been reworded (line 247-250): “Dosing regimens and drug concentrations were chosen based on preliminary studies performed in a mouse model of influenza A virus infection demonstrating antiviral efficacy at 7.2 mg/kg (data not shown), based on the solubility of N-0385 and on knowledge that K18-hACE2 mice typically survive 6 to 8 days post-infection (dpi) with SARS-CoV-2”.

[REDACTED]

[REDACTED]

Future studies by our team will further optimize the therapeutic dose, timing, duration, and frequency of N-0385 and analogues. We are confident that the results presented in this study are sufficient to make the case that peptidomimetic inhibitors targeting TMPRSS2 are an important candidate for treatment and prevention of COVID-19, while recognizing that optimization, including combination therapies, and higher doses will likely further increase this effect. It was beyond the scope of this manuscript to perform optimal dosing regimens of N-0385.

[REDACTED]

[REDACTED]

References

- Beaulieu, A., E. Gravel, A. Cloutier, I. Marois, E. Colombo, A. Desilets, C. Verreault, R. Leduc, E. Marsault, and M. V. Richter. 2013. “Matriptase Proteolytically Activates Influenza Virus and Promotes Multicycle Replication in the Human Airway Epithelium.” *Journal of Virology* 87(8):4237–51.
- Mahoney, Matthew, Vishnu C. Damalanka, Michael A. Tartell, Dong hee Chung, André Luiz Lourenço, Dustin Pwee, Anne E. Mayer Bridwell, Markus Hoffmann, Jorine Voss, Partha Karmakar, Nurit P. Azouz, Andrea M. Klingler, Paul W. Rothlauf, Cassandra E. Thompson, Melody Lee, Lidija Klampfer, Christina L. Stallings, Marc E. Rothenberg, Stefan Pöhlmann, Sean P. J. Whelan, Anthony J. O’Donoghue, Charles S. Craik, and James W. Janetka. 2021. “A Novel Class of TMPRSS2 Inhibitors Potently Block SARS-CoV-2 and MERS-CoV Viral Entry and Protect Human Epithelial Lung Cells.” *Proceedings of the National Academy of Sciences* 118(43):e2108728118.
- Zablockienė, Birutė, Tomas Kačergius, Arvydas Ambrozaitis, Edvardas Žurauskas, Maksim Bratchikov, Laimutė Jurgauskienė, Rolandas Zablockis, and Stefan Gravenstein. 2018. “Zanamivir Diminishes Lung Damage in Influenza A Virus-Infected Mice by Inhibiting Nitric Oxide Production.” *In Vivo* 32(3):473–78.

Reviewer Reports on the Second Revision:

Referee #2 (Remarks to the Author):

The changes to the manuscript answer my previous comments. The clarity in the text helps explain the issues I had previously. Thank you for the changes. The results look great!